# Reduced choice-confidence in negative numerals

**Santiago Alonso-Díaz** ⦿ *, **Gabriel I. Penagos-Londoño**

Department of Economics, Pontificia Universidad Javeriana, Bogotá, Colombia

* alonsosantiago@javeriana.edu.co

## Abstract

Negative numbers are central in math. However, they are abstract, hard to learn, and manipulated slower than positive numbers regardless of math ability. It suggests that confidence, namely the post-decision estimate of being correct, should be lower than positives. We asked participants to pick the larger single-digit numeral in a pair and collected their implicit confidence with button pressure (button pressure was validated with three empirical signatures of confidence). We also modeled their choices with a drift-diffusion decision model to compute the post-decision estimate of being correct. We found that participants had relatively low confidence with negative numerals. Given that participants compared with high accuracy the basic base-10 symbols (0–9), reduced confidence may be a general feature of manipulating abstract negative numerals as they produce more uncertainty than positive numerals per unit of time.

**Data Availability Statement:** Data is available in the Open Science Framework (DOI https://osf.io/yuvz5/).

**Funding:** S.A. received an early career grant from the university (Pontificia Universidad Javeriana. ID

## Introduction

Negative numbers are essential in mathematics. However, they are hard, as exemplified by the reluctance to accept them in the history of math, the trouble of teaching them, their abstract nature, and that humans process them slowly [1–4]. Here we focus on the hypothesis that confidence is worse for negative numerals than positive numerals. Participants picked the larger in a pair of single-digit positive, negative, or 1/positive numerals. Single-digits are important because they are the base-10 symbols and are the foundation for the more convoluted multi-digit processing [5]. To test the hypothesis, we collected implicit measures of confidence (button pressure) and estimated the amount of information produced by negative numerals with a computational model. The possibility of reduced confidence in operations requiring the basic base-10 symbols for inverted numbers (i.e., minus 0–9) is relevant as it could explain learning difficulties via math metacognition [6]. Moreover, studying confidence in single-digits, the primitives of the base-10 system, is important to further our understanding of mathematical cognition and learning of mathematics in children and adults. Before starting, we will use numeral to indicate the symbol and number or magnitude to indicate the mental quantity related to the numeral.

Previous work has focused on the nature of the mental representations of negative numerals. There are two general views on how human cognition represents negative numerals:

PPTA 8329; https://www.javeriana.edu.co/vicerrectoria-de-investigacion/). The funders had no role in study design, data collection and analysis, decision to publish, or preparation of the manuscript.

**Competing interests:** The authors have declared that no competing interests exist.

holistically or strategically. The first one states that negative numerals are assigned a holistic magnitude [4,7,8]. It means that, for instance, when trying to pick the larger between -5 and -3, the brain does so by comparing M(-5) and M(-3). Here the function M is the transformation of the numeral or symbol to an internal mental magnitude.

The second general view, the strategic one, is that there is no representation of negative magnitudes, only of concrete positive quantities. When presented with negative numerals, humans are strategic and flip response or task demands. This hypothesis's main implementation is the component model [1]. Negatives are decomposable into the polarity sign (-) and the positive magnitude. Suppose the task is to pick the larger negative. In that case, the mental comparison is over positive quantities, and the task demand > is changed to <. A more recent iteration of the component model states that negative numbers can be thought as multidigit numbers, with the polarity taking the role of an additional digit that can take only two values (+,-). The presence of a polarity and a digit makes the problem of comparing negative numerals a multi-attribute one [5]. In fact, a neural network implementing multi-attribute decision making can replicate empirical findings such as sign-shortcuts in mixed-comparisons (e.g. when picking the largest between -52 and 34, people just use the fact that there is a negative sign, regardless of the numerical distance between the pair of numerals) and unit-decade compatibility effect where people are faster when the larger (smaller) two-digit number in a pair has at the same time the largest decade and unit (e.g. 42 vs. 57) than when there is an incompatibility between decades and units (e.g. 37 vs. 52) [5]. However, the application of such neural network implementing multi-attribute decision making has only been applied to multidigit comparisons and does not replicate other results such as activation of negative holistic magnitudes in some contexts [8].

Previous work has not adequately addressed the confidence question. This is important because confidence modulates efficient learning [6,9]. Here we use the following definition: confidence is the posterior probability of being correct given a choice and available evidence: p(correct|choice, evidence) [10]. The strategic and holistic hypotheses make similar predictions: negative numbers should elicit less choice confidence. The source, however, is different. The strategic hypothesis predicts that reduced confidence comes from the additional encoding. For instance, dropping the minus sign could induce this confidence: p(correct|choice, drop sign, positive magnitudes, other encoding). However, reduced confidence cannot come from negative magnitude processing because the mind does not represent negative magnitudes, only positive ones. The holistic hypothesis, on the contrary, places part of the reduction of confidence in negative magnitude processing i.e. $\text{confidence}_{pos} = p(\text{correct}|\text{choice, positive magnitudes, other encoding})$ and $\text{confidence}_{neg} = p(\text{correct}|\text{choice, negative magnitudes, other encoding})$.

We will focus on a task that increases the possibility of finding holistic negative magnitudes in the participants. When people compare positive and negative numerals, randomly mixed across trials, response times and accuracy are consistent with the holistic theory [8]. Our goal is the measurement of confidence in negative numbers. One methodological challenge is that people are highly accurate with single digits which could saturate explicit confidence reports. To circumvent this possibility, we collected an implicit motor-based report: button pressure. Previous work in number cognition has found that force is affected by number representations [11–15]. From this work, we know that tasks that do not ask for explicit magnitude comparisons (e.g. parity judgments tasks) induce a categorical response: larger numbers induce strong force and smaller numbers weak force; there is not a smooth gradient by numerical distance [13]. On the other hand, people can map numerical magnitude to a smooth force gradient in tasks that explicitly ask for magnitude estimates (e.g. transform a number into squeezing pressure) [14]. Thus, we argue that measuring button pressure could provide a window into

confidence given that previous literature has found projections of force based on number information.

Here we will ask for a magnitude judgment in a number comparison task where participants must pick the larger number in a pair. Our main interest in this paper is to link button pressure and confidence, and check if numeral type (1/positive numerals, negative numerals, and positive numerals) modulates pressure. In the discussion we will address the consequences for mental representations of numbers and the two conflicting theories for negative numerals (holistic vs component).

Given our interest in confidence, we will confirm three theory-based properties in button pressure [10] (Fig 1): 1) a positive relationship between accuracy and button pressure, 2) higher button pressure in correct trials, and 3) higher accuracy in trials with high button pressure. Thus, if button pressure is a proxy for confidence, it should follow these three characteristics.

The confidence that we study depends on choice and available evidence: p(correct|choice, evidence) [16]. What is critical of this definition is that confidence is an internal estimate that depends on the existence of an actual choice. If there is no choice, there is no confidence under this definition, only uncertainty. In our task, the choice is between a pair of numerals (pick the larger) and the available evidence are the numerals on screen, more precisely, the internal magnitude representation of those numerals. To obtain a proxy of the internal evidence produced by the numerals we simulated the decision process behind single-digit comparisons with a drift diffusion model (DDM) [17]. We will provide details in the following sections. This modeling framework allows simulating an internal decision variable from response times and the accuracy of each participant (Fig 2). More importantly, we can calculate confidence from the internal evidence i.e., from the black trace in Fig 2 (further details in methods).

We found that button pressure follows confidence properties, differed between numeral type, and the decision dynamics obtained with the DDM model was hardly similar between numeral type. Taken together, the results indicate that single-digit negative numerals elicit less confidence. Given the high performance in our simple task (we control for accuracy and response times), the reduction in confidence seems to be a feature of dealing with negative numerals rather than math ability.

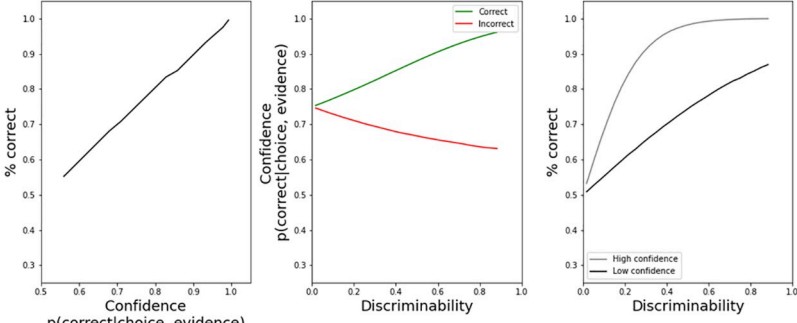

**Fig 1. Illustration of confidence properties.** Correct answers are more likely with higher confidence (left panel), confidence in correct trials is higher (center panel), and, for a given level of discriminability/difficulty, trials with higher confidence should be more accurate (right panel). This figure was simulated using a uniform distribution for discriminability (U(0,1)) and a normal distribution for the perception of discriminability levels (details in [10]). This figure is not a number comparison model (for that, see Fig 5), just an illustration of the confidence properties proposed by [10].

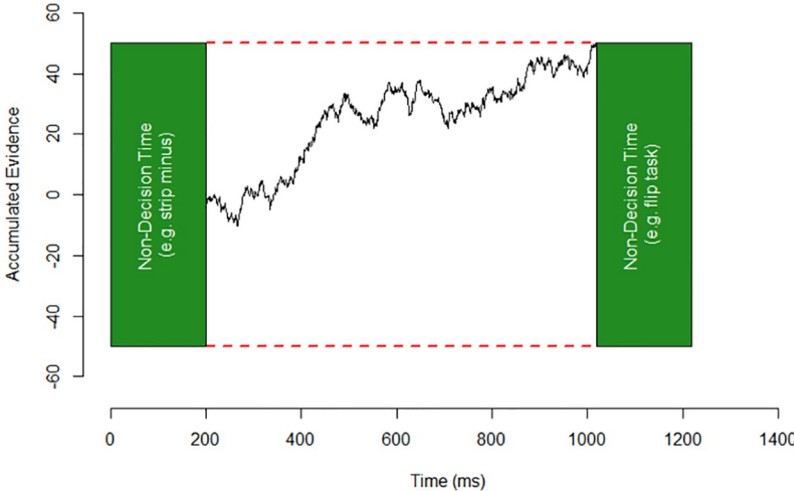

**Fig 2. Decision model.** Individuals select an option when an internal noisy decision variable (black trace) accumulates up to one of the thresholds (red dashes). Non-decision times are constant (green rectangles) and include initial encoding of the stimulus and response-related commands.

## Exp. 1. single-digit numeral comparison task and decision model

Participants had to pick the larger single-digit number in a pair of positives, a pair of negatives, or a pair of 1/positives. The participant saw a random pair type on each trial but never mixed across types (e.g., never a pair containing a positive and a negative). Previous reports have found that mixing trial types induces holistic representations [8] and we are interested in confidence in negative numbers. With Exp. 1 we seek to determine if confidence accrues slower for negative numerals. We do this by using the observed accuracy and response times to model the decision variable leading to a numerosity judgment (Fig 2) and compute confidence levels at the time of choice.

### Materials and methods

*Participants*. We recruited 50 participants for experiment 1 (27 males; mean age: 20.63 years, std.: 1.19). There was no a priori calculation of sample size. However, we will model response times and accuracy, and people are consistently worse with negative numbers than with positive numbers, in studies with sample sizes between 16 to 55 [1,2,4,8]; thus, our sample size is on the larger end. Also, we did a sensitivity analysis to check for the required effect size given an 80% power, p<0.05, a sample size of 50, and a t-test comparing two dependent means. With the G*power application we found that our sample size requires an effect size of 0.4 to achieve that power. This is a medium size effect. Differences in response times between negative and positive numerals in previous literature are approximately 80 to 100 ms (around 10% slower) [1,2,4,8]. The literature we consulted did not report effect sizes but being slower with negative numerals is a highly replicable and easy to find effect. Moreover, the effect sizes in our own data were large and with similar differences as in previous research, between 80 to 120 ms (Cohen's d Exp 1: pos vs neg: 1.68; Exp 2 pos vs neg: 1.46; Exp 3 pos vs neg: 1.96; in the results sections we show the mean and standard deviations, also regressions with estimates and confidence intervals). Thus, we argue that our sample was sufficient to detect response time differences between positive and negative numerals with high power. We emphasize that this sensitivity analysis is a conservative approach given that we did not calculate an a priori sample

size. We used RT for sample size sensitivity as previous research has found that RT correlates with confidence [18].

Participants of Exp. 1 did three tasks probing the processing of inverted information in two different-day sessions lasting 30 to 45 minutes each. The tasks were inverted motion perception, categorization/memorization of inverted items, and numeral comparisons. The order of the tasks was counterbalanced. In this paper, we explain and report the results of the numeral comparison task. Participants were paid approximately 5 U$D in each session (20.000 COPs).

All procedures were in accordance with the ethical standards of the institutional research committee of the economics department of Universidad Javeriana that follows international and national norms regarding research with human subjects. The research committee approved the study with approval code: FCEA-DF-0433. All participants signed a written consent form after it was explained to them.

*Apparatus*. Exp. 1 was run on a 13-inch laptop with a traditional QWERTY keyboard. Stimulus presentation was controlled by Psychtoolbox for Matlab.

*Design*. There were three trial types: a pair of positive, negative, or 1/positive numerals. The type was random on each trial Participants had to pick the larger. The fractional quantities had a simple form (1/positives) as a control. When the numerators is always the same, they are solvable through a denominator strategy [19,20]. Thus, if the comparison of negative numerals is solved by comparing positive values, these two trial types should be more similar than not (e.g., -3 vs. -5 is like 1/3 vs. 1/5, if participants pick the smaller to solve the task correctly). There was not a mixed trial type, say compare a positive to a negative.

We presented numbers from 2 to 15 across 870 trials. 150 of those trials were dummy trials. We called them dummy trials as they are not used for data analysis. We used them so that participants experienced the single digit 9 and double digits. Non-dummy trials (explained below) do not include 1 and 9 to avoid anchor strategies. With anchor strategies we mean that every time 1 or 9 (anchors) are present among single digits, the response is trivial: small (if 1) or large (if 9), without the need of estimating any numerical distance between the pair of digits on-screen. We wanted to avoid such a strategy in our participants given that single-digits do not necessarily and automatically activate magnitude representations [21,22] and strategic behavior is well-known confounder in animal and human choice behavior [23].

For non-dummy trials, namely those that we include in data analysis, we presented single digits using the numbers 2, 3, 5, 7, 8 and generated 1 exemplar pair for each logarithmic numerical distance between the pair (later we compare models with linear or log. num. distances). We did not use 4 and 6 for practical reasons i.e. fewer logarithmic distances (in log. space each pair of digits has a unique distance). Also notice that with the set 2,3,5,7,8 there are the same six possible linear numerical distances as with the set 2,3,4,5,6,7,8. Non-dummy trials did not include the numbers 1 and 9 to avoid anchor strategies (see previous paragraphs for explanation). Participants could not distinguish between dummy and non-dummy trials as they were randomly presented and were otherwise identical.

Participants experimented 10 different exemplars of pairs of digits across 10 logarithmic distances in non-dummy trials. The larger number appeared randomly on the left or right side of the screen. The distribution of non-dummy trials was as follows (Table 1, dummy trials were random, in type and distance, and were not included din any of the analysis).

*Stimuli and procedure*. On every trial, participants saw a pair of numerals on the screen (black background). Their task was to report the larger using the letter Z or M if the left or right was larger, respectively (QWERTY keyboard). The larger appeared randomly on the left or right side of the screen. If a response took more than 3 seconds the trial ended, and the participant saw on-screen a message indicating that the response was too slow. Participants saw a cue to distinguish numeral type in the trial: positive digits were blue, negative digits green, and

**Table 1.**

| log. distance* | Num A | Num B | Trials Exp. 1 | Trials Exp. 2 |
|---|---|---|---|---|
| 0.134 | 8 | 7 | 72 | 36 |
| 0.336 | 7 | 5 | 72 | 36 |
| 0.405 | 3 | 2 | 72 | 36 |
| 0.470 | 8 | 5 | 72 | 36 |
| 0.511 | 5 | 3 | 72 | 36 |
| 0.847 | 7 | 3 | 72 | 36 |
| 0.916 | 5 | 2 | 72 | 36 |
| 0.981 | 8 | 3 | 72 | 36 |
| 1.253 | 7 | 2 | 72 | 36 |
| 1.386 | 8 | 2 | 72 | 36 |

*log(numA)-log(numB).

fractions cyan. When a response was incorrect, the participant received feedback: numbers turned red.

*Data analysis*. We used panel linear regressions to analyze the accuracy (linear probability model) and response times up to two standard deviations from the mean of all the data (i.e., 92% of the trials). The panel regressions included effects for subject-level heterogeneity, i.e., intercept variability by subject.

Specifically, we ran random effects panel regressions to include effects of experiments' order (fixed effects do not allow between-subject variables). We understand random effects as in econometrics [24]: subject-level baselines/intercepts that are independent of the remaining independent variables; a valid assumption because all participants went through the same experimental conditions.

We control for experiment order to check for spill-over effects. Also, we control whether the current trial was preceded by an error to account for behavioral effects caused from transitions from the error feedback.

We included interactions in the regressions when it was theoretically relevant. Namely when the dependent variable was response times (distinct slopes by numeral type suggest different mental magnitudes) and pressure (Fig 1, central panel shows distinct slopes for correct and incorrect trials).

All the regressions use logarithmic distance between the numerals because in all cases it improved the overall model, as measured by BIC (Bayesian information criterion). BIC is a measure of model fit based on the likelihood of the data and a penalty for the number of parameters. In all tables we write the BIC comparison between a model with linear numerical distances and logarithmic distances.

In all regressions, the reference category is positive numerals, hence they do not appear in the tables.

*The drift-diffusion model (DDM)*. We modeled the decision of selecting the larger number in a pair with a drift-diffusion model (Fig 2) [25]. The drift-diffusion model successfully captures accuracy and response times in a myriad of tasks, including number-related ones. It can also describe confidence [26], namely the probability of being correct given a choice and an internal decision variable [16].

In the drift-diffusion framework, the brain produces and accumulates a decision variable at each time step (e.g., millisecond), favoring either of the available options (Fig 2; black trace). One can think of this decision variable as evolving brain activity of areas representing the

relevant cognitive variables. In more abstract terms, the decision variable is all the information necessary to produce a choice.

In a numeral comparison task, the decision variable is the perceived distance between the pair of numerals [27,28]. This decision variable is noisy, so it must accumulate to a threshold before committing to an option. For example, suppose the internal decision variable hits a threshold A, representing the largest numeral. In that case, the participant selects option A. However, if it hits threshold B, they select option B, representing the incorrect smaller numeral. Thus, incorrect decisions exist because the decision variable is noisy and there is a probability of arriving at the wrong answer.

Threshold levels, the speed of accumulation (drift rate), noise, and starting point of the decision variable, are considered decision parameters as they determine the evolution of the black trace in Fig 2. The model also considers constant non-decision times (Fig 2; green squares). These are times unrelated to the formation and accumulation of the decision variable. They include initial encoding (e.g., if needed, strip minus signs) and response-related commands (e.g., if needed, flip task from < to >; or flip the left index to the right index finger to execute the motor command). The time it takes to arrive to either threshold plus the constant non-decision times is the response time.

An underlying assumption of the framework is that decision and non-decision times are independent and sequential; multi-step decision-variables are not part of the standard drift-diffusion theory [17]. Thus, our modeling exercise assumes that numerals' mental magnitudes are unrelated to the numerals' initial encoding and final response commands i.e. in Fig 1 black trace and green squares are independent. In other words, we assume that stimulus and response encoding (Fig 2, green square), such as detecting if there is a negative numeral on-screen, stripping the minus sign, or pressing the left or right key, are independent of the main decision loop based on noisy estimates of numerical distance (Fig 2, black trace). Independence between encoding and magnitude processing is not unusual in the literature. For instance, in the component model, polarity information (plus or minus) is separated from magnitude information [1]. In the holistic theory of [4], they propose the formula $-B^*M(|x|)$ for negative numeral magnitudes. Here x is the numeral, M is a function that transforms the numeral to a mental magnitude, and B implements compression and placing the magnitude on the negative section of the mental number line (the minus sign). Note that M does not take the flipping operation -B as an argument i.e. M is independent of non-magnitude processing.

The drift-diffusion model (Fig 2) had six parameters: 1) drift rate of evidence (DR), 2) symbolic manipulation (SYM), 3) ± Bound (BO), 4) range around zero for the inter-trial variability of the initial point of accumulation (IC), 5) non-decision times (NDT), and 6) inference uncertainty (UN). The drift rate determines how fast information is accumulated and depends on trial difficulty such that easy trials accrue information faster: $DR_{trial} = DR^*Num.\ distance^{Sym}$. The parameter SYM accounts for the possibility that symbols enhance mental magnitudes manipulation [29]. It can take any value between 0 (symbols cancel numerical distance effects) and 1 (raw numerical distance affects evidence drift). For numerical distance, we used logarithmic scales [30]. Inter-trial variability of the initial point of accumulation (IC) is a range around zero and non-decision times (NDT) is a constant time added to the obtained decision time on each trial.

The parameter UN determines the precision of a Bayesian inference on the mean value of the decision variable ($\mu_{dv}$; black trace in Fig 2). Specifically, by Bayes rule:

$$p(\mu_{dv}|dv) \propto p(dv|\mu_{dv})p(\mu_{dv})$$

With a normal likelihood ($p(dv \mid \mu_{dv})$) and an uniform prior $p(\mu_{dv})$, the resulting posterior is also normal (for derivation see, [31]):

$$p(\mu_{dv}|dv) \sim Normal\left(\overline{dv}, \frac{UN}{\sqrt{n}}\right) \qquad (1)$$

The mean ($\overline{dv}$) is the mean of the decision-variable stream up to the sample n. The standard deviation is the inference uncertainty divided by the square root of the total samples n: $UN/\sqrt{n}$. Large UN mean that more samples are needed to reduce uncertainty on the estimated $\mu_{dv}$.

Given that positive values of the decision variable (dv) indicate a correct response, we estimate confidence with the cumulative probability that $\mu_{dv}$ is greater than zero at threshold. Specifically, $p(\mu_{dv}|dv) > 0$. It is important to highlight that uncertainty UN is not the same as confidence in the model. UN is the base standard deviation of the inference, while confidence is the cumulative probability that $\mu_{dv}$ is greater than zero after a choice is made (i.e. after dv arrives at one of the thresholds).

Drift rate (DR), compression due to symbolic manipulation (SYM), Bound (BO), inter-trial variability of the initial point of accumulation (IC), and non-decision times (NDT) were fitted to each individual data using the pyDDM library for Python [32], and after excluding slow response times ($> 2$ std. dev from the mean of all the data; we lost approx. 8% of trials). Each numeral type was fitted separately; thus, we assume that non-decision times captures encoding differences, such as detecting whether there is a negative, positive, or 1/positive numeral on screen or dropping the minus sign. Once encoded, each numeral type generates its own decision process.

To check if drift changed between positive and negative numerals, we calculated $|DR_{pos} - DR_{neg}|$. With the resulting vector, we did a one-sample t-test against zero change (p-values corrected for multiple comparisons with Holm-Sidak). We used the absolute distances as we do not have a particular directional hypothesis (but see paired t-tests in Supplemental Information).

To implement Eq 1, we simulated multiple trials for each participant (150 per num. distance) and with the resulting decision-variable stream at the end of each trial we applied Eq 1. We manually set inference uncertainty (UN) to qualitative demonstrate confidence effects. UN does not affect the fitting procedure; it is a free parameter.

## Results

Mean accuracy was almost at ceiling (Fig 3; Exp. 1 (mean,std): 1/pos: 0.95, 0.04 neg: 0.93, 0.05, pos: 0.92, 0.05). Table 1.1 reveals that accuracy was slightly higher with negative numerals and 1/n numerals. Accuracy improved with larger numerical distances. Experiment order in Exp. 1 was a significant predictor of accuracy (but still high regardless of order i.e. intercept 88%). Those that started with the random dot motion task on day 1 and then on day 2 did the numeral and memory task (i.e. order RDM_N), had better accuracy. If a trial was preceded by an error, it was more likely to be correct on that trial.

Response times behaved similarly as in previous number cognition research (Table 1.2; Exp. 1 (mean,std): 1/pos: 775 ms, 101 ms; neg: 792 ms, 94 ms; pos: 710 ms, 83 ms). Participants were slower with negative and 1/n numerals. Also, there was an effect of numerical distance. This means that as the distance between numerals increased response times got faster (Table 1.2). The RT slopes for fractions and negatives were steeper (Table 1.2, interaction terms with distance). This difference is consistent with the holistic theory. The strategic hypothesis does not predict changes in slope/processing speed because we do not represent

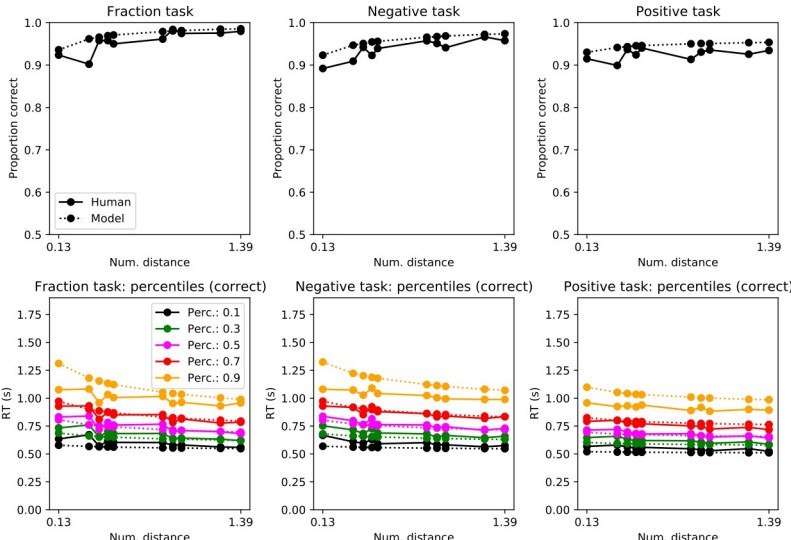

**Fig 3. Accuracy (top) and response time distributions by percentiles (bottom) by trial type (Exp 1.).** Numerical distance in log. scale. Response times are for correct trials; see Supplemental Information for incorrect trials.

negative magnitudes. Participants who experienced the order RDM_N were faster. If a trial was preceded by an error, that trial was slower.

In summary, the behavioral results replicate traditional outcomes in the number cognition literature: high accuracy in single-digit comparisons, distance effects in response times, and slower processing of negative numerals.

Given the observed accuracy and response times we can model the decision process. The decision model simulates cognitive evidence and we can compute a confidence metric to see if negative and positive numerals produce distinct confidence levels. We simulate the decision process with a drift diffusion model and illustrate with this framework that, given the observed response times and accuracies, positive numerals generate stronger internal evidence per unit

**Table 1.1. Accuracy.** Exp. 1.1 Random Effects Estimation.

|  | Par. | Std.Err | t | p | Low CI | High CI |
|---|---|---|---|---|---|---|
| Intercept | 0.88 | 0.01 | 127.04 | 0 | 0.87 | 0.89 |
| 1/n | 0.03 | 0 | 9.57 | 0 | 0.02 | 0.04 |
| Neg. | 0.01 | 0 | 3.59 | 0 | 0.01 | 0.02 |
| Order M_RDM | 0.01 | 0 | 3.18 | 0 | 0 | 0.02 |
| Order RDM_N | 0.05 | 0.01 | 8.64 | 0 | 0.04 | 0.06 |
| Order N_RDM | 0.02 | 0 | 4.71 | 0 | 0.01 | 0.02 |
| Num. dist | 0.04 | 0 | 10.84 | 0 | 0.03 | 0.04 |
| RT | 0.01 | 0.01 | 1.3 | 0.19 | 0 | 0.02 |
| Error_next_trial | 0.06 | 0 | 42.82 | 0 | 0.06 | 0.07 |
| Cov. Estimator: | Robust | Log-likelihood | 784.43 | F (8,32907): | 236.28 | |
| No. subj: | 50 | No. Obs: | 32916 | P-value | 0 | |
| BIC: | -1475 | BIC vs Linear: | -45 | | | |

M: Memory task; RDM: Random dot motion task; N: Numeral task,

**Table 1.2. RT. Exp. 1**. Random Effects Estimation.

|  | Par. | Std.Err | t | p | Low CI | High CI |
|---|---|---|---|---|---|---|
| Intercept | 0.75 | 0.02 | 39.66 | 0 | 0.71 | 0.78 |
| 1/n | 0.1 | 0.01 | 19.19 | 0 | 0.09 | 0.11 |
| Neg. | 0.1 | 0.01 | 19.51 | 0 | 0.09 | 0.11 |
| Order M_RDM | 0 | 0.03 | 0.02 | 0.98 | -0.05 | 0.05 |
| Order RDM_N | -0.14 | 0.06 | -2.18 | 0.03 | -0.26 | -0.01 |
| Order N_RDM | -0.01 | 0.03 | -0.27 | 0.79 | -0.06 | 0.05 |
| Num. dist | -0.04 | 0 | -10.44 | 0 | -0.05 | -0.04 |
| 1/n:Dist | -0.05 | 0.01 | -7.66 | 0 | -0.06 | -0.03 |
| Neg:Dist | -0.03 | 0.01 | -4.12 | 0 | -0.04 | -0.01 |
| Error_next_trial | 0.02 | 0 | 4.34 | 0 | 0.01 | 0.03 |
| Cov. Estimator: | Robust | Log-likelihood | 9347.6 | F (9,32906): | 223.87 |  |
| No. subj: | 50 | No. Obs: | 32916 | P-value | 0 |  |
| BIC: | -18591 | BIC vs Linear: | -275 |  |  |  |

M: Memory task; RDM: Random dot motion task; N: Numeral task.

of time. The drift diffusion model, as any modelling exercise, has assumptions (see Methods), but given the framework and assumptions it provides insights into human confidence in numerals.

**Drift diffusion results.** The drift-diffusion decision model captured mean accuracy and response time percentiles (Fig 3). However, the 0.9 percentile was faster in participants. Such a faster response in the 0.9 percentile could indicate other processes unrelated to the task that the model does not capture. Given that this percentile is the slowest, perhaps the response happened before ending cognitive computations.

To account for the high accuracy, we introduced a symbolic facilitation/compression parameter. We tried a simpler model without it but it failed to produce high accuracies (i.e. worse fit). Thus, in single-digit symbolic tasks, there seems to be facilitation by symbols that boost accuracy, consistent with the notion that numerals are a cognitive technology [29].

There was a change in decision dynamics across inverted (negative and 1/positive) and positive numerals (Table 1.3). This change is not concentrated in non-decision times, suggesting that it is not a simple encoding effect (e.g. drop signs). Drift, symbolic compression, bounds, and range of initial point of accumulation differed. We tried paired-sample t-tests to test directionality but it was not evident (Supplemental Information). This could mean that there is not a universal effect on how numeral type affects decision making parameters; some participants may modulate their drift rate, others compensate by reducing decision bounds; others strictly follow an encoding strategy such as dropping minus signs, and so on.

In the model, confidence is defined as the probability that the decision variable was positive during all the trial (i.e. the black trace in Fig 2). We calculated confidence at the end of each trial, namely when the decision variable hit one of the thresholds i.e. when a choice was made. The model has the expected characteristics (Fig 4): a) accuracy improves with higher confidence, b) correct trials have large confidence, and c) trials with larger confidence, as determined by a median cut, are more accurate.

The free parameter affecting confidence in the model was the uncertainty parameter UN (Eq 1). We highlight that UN is not the same as confidence; uncertainty UN is the parameter modulating the standard deviation of the inference on the accumulated evidence (Eq 1), while confidence is the probability of being correct after committing to a choice. We tested two

**Table 1.3. Avg. parameters of individual fits and one-sample t-tests (vs zero change).**

*Experiment 1*

|  | Drift | Sym. Compress | Bound | IC_range | NDT |
|---|---|---|---|---|---|
| **1/n** |  |  |  |  |  |
| mean | 3.01 | 0.22 | 1.03 | 0.32 | 0.43 |
| std | 0.88 | 0.11 | 0.65 | 0.21 | 0.09 |
| t-test vs. neg | t(49) = 8.37 | t(49) = 8.81 | t(49) = 2.90 | t(49) = 8.21 | t(49) = 4.72 |
| t-test vs. pos | t(49) = 9.75 | t(49) = 10.71 | t(49) = 3.04 | t(49) = 7.05 | t(49) = 5.31 |
| **Negative Num.** |  |  |  |  |  |
| mean | 2.56 | 0.19 | 0.92 | 0.33 | 0.43 |
| std | 0.71 | 0.1 | 0.2 | 0.18 | 0.07 |
| t-test vs. pos | t(49) = 8.58 | t(49) = 9.92 | t(49) = 7.73 | t(49) = 7.57 | t(49) = 9.70 |
| **Positive Num.** |  |  |  |  |  |
| mean | 2.46 | 0.09 | 0.77 | 0.22 | 0.41 |
| std | 0.77 | 0.12 | 0.13 | 0.09 | 0.06 |

*Experiment 2*

|  | Drift | Sym. Compress | Bound | IC_range | NDT |
|---|---|---|---|---|---|
| **1/n** |  |  |  |  |  |
| mean | 3 | 0.14 | 1.59 | 0.37 | 0.69 |
| std | 1.04 | 0.13 | 1.77 | 0.27 | 0.2 |
| t-test vs. neg | t(46) = 6.92 | t(46) = 8.21 | t(46) = 2.71 | t(46) = 8.78 | t(46) = 3.64 |
| t-test vs. pos | t(46) = 6.22 | t(46) = 8.92 | t(46) = 3.22 | t(46) = 6.12 | t(46) = 4.73 |
| **Negative Num.** |  |  |  |  |  |
| mean | 2.79 | 0.14 | 1.32 | 0.45 | 0.72 |
| std | 0.85 | 0.13 | 0.77 | 0.23 | 0.16 |
| t-test vs. pos | t(46) = 7.62 | t(46) = 10.27 | t(46) = 4.40 | t(46) = 8.80 | t(46) = 5.64 |
| **Positive Num** |  |  |  |  |  |
| mean | 2.65 | 0.09 | 1.2 | 0.3 | 0.67 |
| std | 0.91 | 0.14 | 0.76 | 0.18 | 0.16 |

*Experiment 3*

|  | Drift | Sym. Compress | Bound | IC_range | NDT |
|---|---|---|---|---|---|
| **1/n** |  |  |  |  |  |
| mean | 2.93 | 0.16 | 1.27 | 0.40 | 0.67 |
| std | 0.85 | 0.12 | 0.61 | 0.27 | 0.12 |
| t-test vs. neg | t(44) = 8.96 | t(44) = 8.69 | t(44) = 4.43 | t(44) = 7.27 | t(44) = 5.63 |
| t-test vs. pos | t(44) = 8.13 | t(44) = 9.31 | t(44) = 5.19 | t(44) = 7.55 | t(44) = 7.18 |
| **Negative Num.** |  |  |  |  |  |
| mean | 2.49 | 0.14 | 1.12 | 0.49 | 0.71 |
| std | 0.61 | 0.14 | 0.26 | 0.23 | 0.14 |
| t-test vs. pos | t(44) = 8.74 | t(44) = 8.61 | t(44) = 7.73 | t(44) = 9.27 | t(44) = 5.83 |
| **Positive Num.** |  |  |  |  |  |
| mean | 2.31 | 0.11 | 0.88 | 0.23 | 0.66 |
| std | 0.55 | 0.12 | 0.15 | 0.10 | 0.08 |

All p-values < 0.05, and corrected for multiple (15) comparisons (Holms-Sidak).

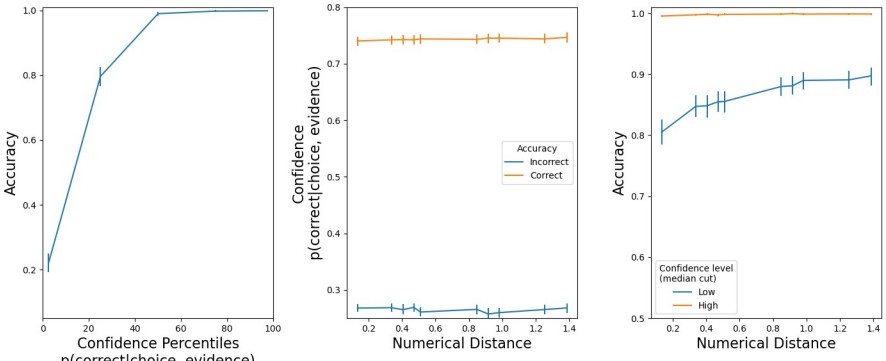

**Fig 4. Confidence properties in the drift diffusion model (Exp 1).** Left panels: accuracy increased with more confidence. Center panels: there was low confidence in incorrect trials. Right panels: accuracy is higher in high confidence trials. Error bars are s.e.m.

possibilities. First, uncertainty UN was equal for 1/positive, negative, and positive numbers. Second, uncertainty UN was lower for positive numbers. This second option represents the possibility that positive numbers generate more information per sample. In Eq 1, the standard deviation is divided by the square root of the number of samples. Therefore, if UN is lower, the standard deviation of the inference gets tighter faster and will improve confidence with fewer samples.

Fig 5 shows that if we assume equal uncertainty UN, trials with positive numbers produce less confidence (center panels, Exp. 1, 2, and 3). This happens because trials with positive numbers are generally faster (Table 1.2), producing fewer samples, and an estimate based on Eq 1 is less confident. On the other hand, if we assume that positive trials have a lower uncertainty for each sample of the decision variable, as indexed by the UN parameter, then positive numbers increase in confidence (right panels).

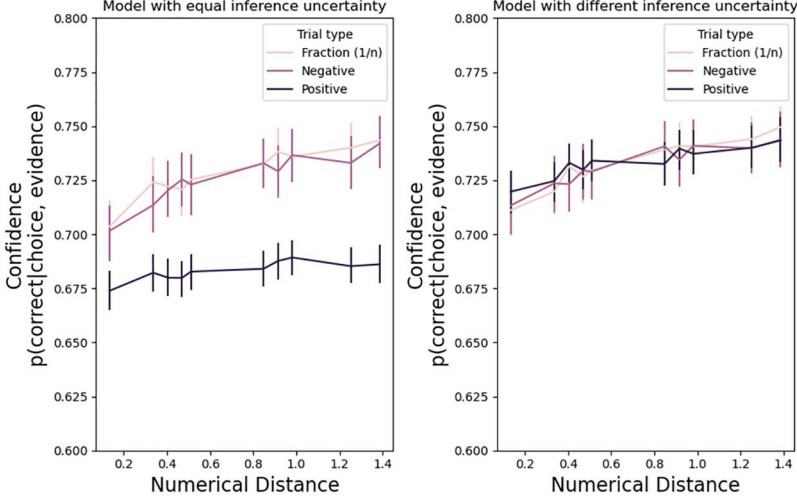

**Fig 5. Reduced uncertainty in trials with positive numerals.** Exp. 1. models with equal uncertainty had an UN = 2.5 for all types of numerals. Models with different uncertainty had $UN_{pos}$ = 1.65, $UN_{neg}$ = 2.4, $UN_{frac}$ = 2.4. Error bars are s.e.m.

To reiterate, this qualitative result from the model suggests two possibilities for confidence in negative numerals judgments: 1) negative numerals induce *more confidence* because they have more samples at choice and they have similar uncertainty UN as positive numerals (note that Fig 5 already takes into account potential difference in the other parameters of the model); or, on the other hand, 2) negative numerals could induce *less confidence* because their uncertainty UN is larger.

## Exp. 2 confidence in single-digit comparisons

In Exp. 2 we collected an implicit measure of confidence: button pressure. We assumed that more substantial button pressure indicates confidence. Below we provide a data-based confirmation of this assumption. Importantly, the overall results of Exp. 2 suggest that negative numerals induce less confidence. This means that negative numerals produce less certainty per unit of time i.e. higher uncertainty parameter UN.

### Materials and methods

*Participants*. We recruited 49 participants for experiment 2 (24 males; mean age: 20.78 years, std: 1.36; two participants' data was saved incorrectly, and one subject mistakenly used the keyboard instead of the force sensors because we presented, by mistake, instructions indicating the available response keys on the keyboard; final sample for regressions evaluating pressure n = 46; final sample for drift diffusion modelling n = 47). There was no a priori calculation of sample size but in the preregistration of Exp. 2 we explicitly limited the number of participants to 50.

Participants of Exp. 2 did two tasks the same day: inverted motion perception and numeral comparisons. The order of the tasks was counterbalanced. In this paper, we explain and report the results of the numeral comparison task. Participants were paid approximately 5 U\$D in each session (20.000 COPs).

All procedures were in accordance with the ethical standards of the institutional research committee of the economics department of Universidad Javeriana that follows international and national norms regarding research with human subjects. The research committee approved the study with approval code: FCEA-DF-0433. All participants signed a written consent form after it was explained to them.

*Apparatus*. Exp. 2 was run on a 13-inch laptop. Stimulus presentation was controlled by Psychtoolbox for Matlab. We used two-force sensors below each index finger (Force Sensitive Resistor Interlink 402; 10kΩ resistor; see diagram of circuit in Supplemental Information). The force sensitive resistor changes its resistance as a function of pressure and a microcontroller (Arduino UNO) produces values between 0 and 1023. It detects pressure as a change in resistance, not weight. Participants pressed one of the force sensors to report their decision on each trial. The microcontroller relayed information to Matlab, which presented the stimuli using Psychtoolbox. The force sensors produced a continuous pressure signal during a trial (see example trials in Supplemental Information). The Arduino's baud rate was 9600 bits per second. We transmitted 24 characters (240 bits) and 4 floats (128 bits), for a sampling rate of 26 Hz (i.e. 9600/368). Specifically, the Arduino sent four strings of four characters, eight next lines (via Serial.println in the Arduino code), and four float numbers, to Matlab approximately every 38 milliseconds. However, empirically, we observed that Matlab collected this information around every 45 to 50 milliseconds on average, perhaps due to other processing delays (e.g. buffering, data setup, and stimulus presentation).

*Stimuli, procedure, and design*. The procedure and design were like Exp. 1 but with fewer trials (435 + 75 dummy trials). The main difference was that we captured button pressure for each response.

*Data analysis*. Experiment 2 followed a similar analysis as Exp. 1 (panel linear regressions with random effects and the drift diffusion model). It was preregistered in https://osf.io/gqtja. However, this work has evolved thanks to exposure in conferences and journals. We added the drift-diffusion model and panel regressions. Thus, the preregistration confirms that we did not change the hypothesis after obtaining the results (i.e., no HARCKing; we build the whole button pressure apparatus to test the hypothesis) but the analytic approach did change. We hope that this clarification makes the life of research reports more transparent under a preregistration model of science.

We analyzed the maximum pressure on each trial, standarized to the maximum pressure during the whole task of each participant (separately for the left and right sensor). For instance, if subject X max pressure during his whole session was 984 on the left sensor, then all left sensor pressures during the session were divided by 984 and we used for an specific trial the max value. We used this peak value in the regressions. The observed range for this dependent variable, after excluding slow response times (> 2 std. dev from the mean; we lost approx. 8% of trials), was for Exp. 2 [0.025, 1] and for Exp. 3 [0.05, 1]. We used pressure signals (see Supplemental Information) in the following interval: as soon as the numerals appeared on screen and until the subject selected an option and was no longer pressing the force sensor (i.e. force resistance < low_pressure_threshold of 100).

## Results

Mean accuracy was almost at ceiling (Fig 6; Exp 2 (mean,std): 1/pos: 0.97, 0.02 neg: 0.96, 0.03 pos: 0.95, 0.04). Table 2.1 reveals that accuracy was slightly higher with negative numerals and 1/n numerals. Accuracy improved with larger numerical distances. Experiment order in Exp. 2 was not a significant predictor of accuracy (and high regardless of order i.e. intercept 90%). If a trial was preceded by an error, it was more likely to be correct on that trial.

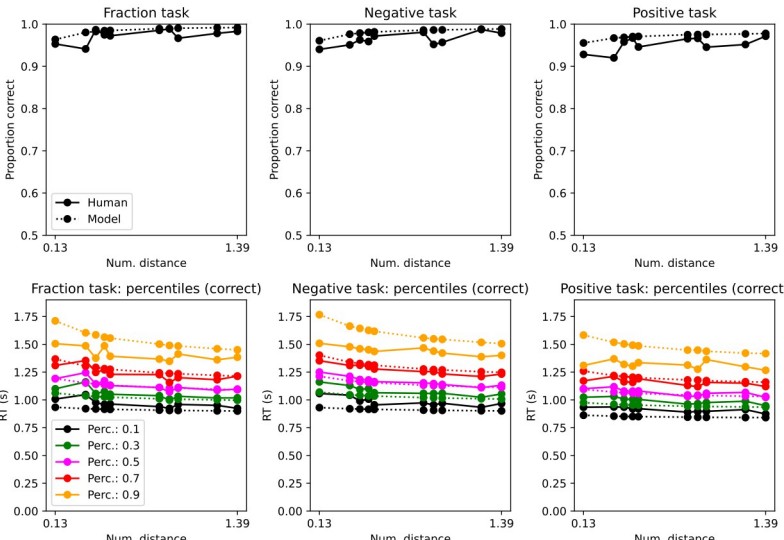

**Fig 6. Accuracy (top) and response time distributions by percentiles (bottom) by trial type (Exp 2.).** Numerical distance in log. scale. Response times are for correct trials; see Supplemental Information for incorrect trials.

**Table 2.1. Accuracy. Exp. 2**. Random Effects Estimation.

|  | Par. | Std.Err | t | p | Low CI | High CI |
|---|---|---|---|---|---|---|
| Intercept | 0.9 | 0.01 | 78.64 | 0 | 0.88 | 0.92 |
| 1/n | 0.02 | 0 | 5.81 | 0 | 0.01 | 0.03 |
| Neg. | 0.01 | 0 | 3.25 | 0 | 0.01 | 0.02 |
| Max. press | 0.05 | 0.01 | 5.36 | 0 | 0.03 | 0.07 |
| Num. dist | 0.02 | 0 | 5.54 | 0 | 0.01 | 0.03 |
| RT | 0 | 0.01 | -0.32 | 0.75 | -0.02 | 0.01 |
| Order | 0 | 0 | -1.26 | 0.21 | -0.01 | 0 |
| Error_next_trial | 0.04 | 0 | 21.49 | 0 | 0.03 | 0.04 |
| Cov. Estimator: | Robust | Log-likelihood | 3711.1 | F (7,14920): | 69.67 | |
| No. subj: | 46 | No. Obs: | 14928 | P-value | 0 | |
| BIC: | -7345 | BIC vs Linear: | -10 | | | |

For experiment 2 we aimed to confirm the presence of confidence properties in button pressure and if numeral type modulated such pressure. We present three regressions, one for each panel of the confidence theory represented in Fig 1, while controlling for response times.

Accuracy and confidence should have a positive relation (Fig 1, left panel). Table 2.1 confirms this for Exp. 2. The pressure estimate is positive meaning that trials with higher pressure were more likely to be correct. Moreover, a simple regression, just including button pressure as a regressor, to directly test the theoretical confidence property in the first panel in Fig 1, also finds a positive relation (Exp. 2: $\beta$ = 0.05, 95%CI = [0.03, 0.07], p <0.01).

For a given discriminability (i.e. numerical distance) trials with high confidence, as defined by a median split, should be more accurate (Fig 1, right panel). Table 2.2. reveals that a regressor for a dummy for the median split of max. button pressure is significant. This means that when subjects were correct, they pressed the force sensor harder, controlling for numerical distance. Moreover, a simple regression, just including the median split and numerical distance as regressors, to directly test the theoretical confidence property in the third panel in Fig 1, also finds a positive estimate for the median split (Exp. 2: $\beta$ = 0.01, 95%CI = [0, 0.02], p <0.01).

Correct trials should have higher levels of confidence than incorrect trials for a given discriminability (Fig 1, central panel). Table 2.3 shows that indeed correct trials have higher button pressure (correct estimate), in a regression that controls for numerical distance.

**Table 2.2. Accuracy. Exp. 2**. Random Effects Estimation.

|  | Par. | Std.Err | t | p | Low CI | High CI |
|---|---|---|---|---|---|---|
| Intercept | 0.93 | 0.01 | 92.15 | 0 | 0.91 | 0.95 |
| 1/n | 0.02 | 0 | 5.66 | 0 | 0.01 | 0.03 |
| Neg. | 0.01 | 0 | 3.04 | 0 | 0 | 0.02 |
| Median max. press. (dummy) | 0.01 | 0 | 3 | 0 | 0 | 0.02 |
| Num. dist | 0.02 | 0 | 5.62 | 0 | 0.01 | 0.03 |
| RT | 0 | 0.01 | 0.46 | 0.64 | -0.01 | 0.02 |
| Order | 0 | 0 | -1.35 | 0.18 | -0.01 | 0 |
| Error_next_trial | 0.04 | 0 | 22.02 | 0 | 0.03 | 0.04 |
| Cov. Estimator: | Robust | Log-likelihood | 3695.6 | F (7,14920): | 73.111 | |
| No. subj: | 46 | No. Obs: | 14928 | P-value | 0 | |
| BIC: | -7314 | BIC vs Linear: | -11 | | | |

**Table 2.3. Max. Press.** Exp 2. Random Effects Estimation.

|  | Par. | Std.Err | t | p | Low CI | High CI |
|---|---|---|---|---|---|---|
| Intercept | 0.56 | 0.02 | 22.66 | 0 | 0.52 | 0.61 |
| 1/n | -0.01 | 0 | -3.61 | 0 | -0.02 | -0.01 |
| Neg. | -0.02 | 0 | -4.75 | 0 | -0.02 | -0.01 |
| Correct | 0.04 | 0.02 | 2.13 | 0.03 | 0 | 0.07 |
| Num. dist | -0.01 | 0.02 | -0.62 | 0.54 | -0.06 | 0.03 |
| Dist:Correct | 0.02 | 0.02 | 0.75 | 0.45 | -0.03 | 0.06 |
| RT | 0.1 | 0.01 | 15.72 | 0 | 0.09 | 0.12 |
| Order | 0 | 0.02 | -0.18 | 0.86 | -0.05 | 0.04 |
| Error_next_trial | 0 | 0.01 | 0.15 | 0.88 | -0.01 | 0.02 |
| Cov. Estimator: | Robust | Log-likelihood | 4946 | F (8,14919): | 33.992 | |
| No. subj: | 46 | No. Obs: | 14928 | P-value | 0 | |
| BIC: | -9806 | BIC vs Linear: | -2 | | | |

Moreover, a simple regression, excluding numeral types and just including a dummy for correct trials and numerical distance, to directly test the theoretical confidence property in the central panel in Fig 1, further confirms this overall higher button pressure in correct trials (Exp. 2: $\beta = 0.05$, 95%CI = [0.03, 0.06], p<0.01). Fig 1, central panel, shows distinct slopes for correct and incorrect trials (positive and negative). However, the interaction term was not significant, but it was positive in line with the theory. Our participants had a low error rate and could explain the lack of interaction.

Fig 7 has a visualization of all the confidence properties in max. button pressure that we explained in the previous paragraphs. Accuracy was higher with stronger button pressures (left panel). Button pressure was higher in correct trials (center panel). For a given difficulty, accuracy was higher in trials with higher confidence (right panel). The presence of these three properties indicates that participants expressed their confidence level with button pressure.

Importantly, in Exp. 2 trial type affected button pressure (Table 2.3). Participants reduced button pressure in trials with 1/positive and negative numerals. The reduction of button pressure in inverted numerals (negative and 1/n numerals) was present in the regressions shown in Table 2.3; they are not significant as simple main effects without controlling for the other variables.

Response times behaved similarly as in previous number cognition research (Table 2.4; Exp. 2 (mean,std): 1/pos: 1170 ms, 178 ms; neg: 1197 ms, 183 ms; pos: 1098 ms, 149 ms). Participants were slower in inverted trials (negative and 1/n numerals). There was an effect of numerical distance. As the distance between numerals increased response times got faster (Table 2.4). The RT slopes for fractions and negatives were steeper (Table 2.4, interaction terms with distance). This difference is consistent with the holistic theory. The strategic hypothesis does not predict changes in slope/processing speed because we do not represent negative magnitudes.

In summary, the empirical observations indicate that button pressure is a proxy for confidence, even after controlling for response times and other confounders. Importantly, participants seem to be more confident when comparing a pair of positive numerals than the other type of numerals.

In the following section we simulate the decision process with a drift diffusion model and illustrate with this framework that, given the observed response times and accuracies, positive numerals generate stronger internal evidence per unit of time. Also, the modelling exercise

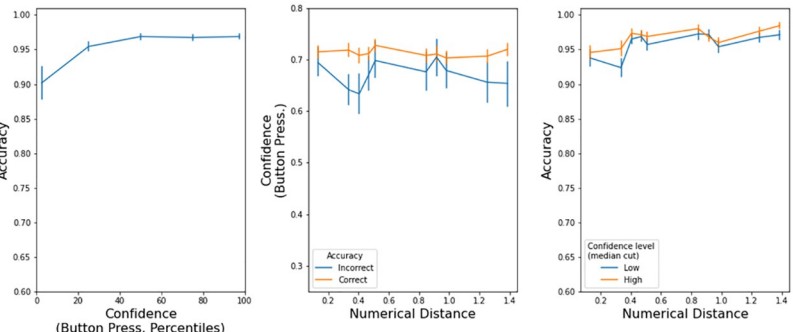

**Fig 7. Confidence characteristics in button pressure (Exp. 2).** Error bars are s.e.m.

explains why response times correlate positively with confidence (Table 2.3, RT estimate): more information increases confidence by reducing, with each sample, the standard deviation of the probability distribution of being correct (Eq 1).

**Drift diffusion results.** As with Exp. 1., the drift-diffusion decision model captured mean accuracy and response time percentiles (Fig 6). Also, there was a change in decision dynamics across inverted (negative and 1/positive) and positive numerals (Table 1.3).

The model has the expected characteristics (Fig 8): a) accuracy improves with higher confidence, b) correct trials have large confidence, and c) trials with larger confidence, as determined by a median cut, are more accurate. The model is quantitatively different from the observed button pressure because we do not know (so we could not implement) a transfer function Confidence -> Button pressure. Still, the confidence output of the drift-diffusion model is insightful because, given the observed response times and accuracy, we can look at how confidence behaves under the computational model.

As with Exp. 1, Fig 9 shows that if we assume equal uncertainty UN, trials with positive numbers produce less confidence (center panels). However, on the left panels, we present actual button pressure. Button pressure for positive numbers is not weaker, if anything stronger (Table 2.3). Thus, a model that assumes lower information uncertainty for positive numbers is qualitatively better at reflecting the observed effects of button pressure across numeral types. Given the observed accuracy and response times used to estimate the DDM parameters, negative numerals seem to induce a higher uncertainty UN.

**Table 2.4. RT. Exp. 2**. Random Effects Estimation.

|  | Par. | Std.Err | t | p | Low CI | High CI |
|---|---|---|---|---|---|---|
| Intercept | 1.17 | 0.03 | 38.67 | 0 | 1.11 | 1.23 |
| 1/n | 0.1 | 0.01 | 10.03 | 0 | 0.08 | 0.12 |
| Neg. | 0.12 | 0.01 | 12.66 | 0 | 0.1 | 0.14 |
| Num. dist | -0.05 | 0.01 | -6.79 | 0 | -0.07 | -0.04 |
| 1/n:Dist | -0.03 | 0.01 | -2.92 | 0 | -0.06 | -0.01 |
| Neg:Dist | -0.03 | 0.01 | -2.46 | 0.01 | -0.05 | -0.01 |
| Order | -0.04 | 0.04 | -1.04 | 0.3 | -0.11 | 0.04 |
| Error_next_trial | 0.04 | 0.01 | 3.5 | 0 | 0.02 | 0.06 |
| Cov. Estimator: | Robust | Log-likelihood | 579.26 | F (7,14920): | 110.96 | |
| No. subj: | 46 | No. Obs: | 14928 | P-value | 0 | |
| BIC: | -1082 | BIC vs Linear: | -88 | | | |

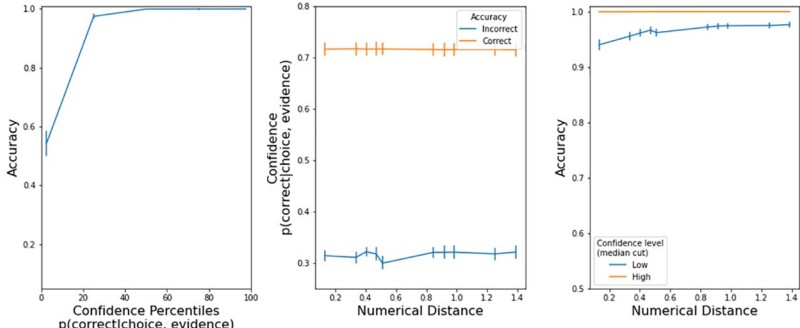

**Fig 8. Confidence properties in the drift diffusion model (Exp 2).** Left panels: accuracy increased with more confidence. Center panels: there was low confidence in incorrect trials. Right panels: accuracy is higher in high confidence trials. Error bars are s.e.m.

## Exp. 3 replication of the effects with only one experimental session, no color cues, and no feedback

In Exp. 1 and 2, participants did more tasks, related to inversion of information (e.g. report the anti-direction of moving dots). Even though we controlled for order effects, it is important to fully address this concern. In Exp. 3 participants only did the single-digit numeral comparison task. Also, Exp. 1 and 2 provided error feedback and this could affect responses. In Exp. 3 no such feedback appears. Finally, in Exp. 1 and 2 numeral types had different colors i.e. positive blue, negatives green, 1/n cyan. In Exp. 3 we drop such color cues.

### Materials and methods

*Participants.* We recruited 50 participants for Experiment 3 (19 males; mean age: 19.44 years, std: 2.18; one participant's data was saved incorrectly and four had an error rate larger than 15%, unusual for single digit comparisons, final n = 45; in Supplemental Information we present analyses including the four participants with large error rate and the results are similar). There was no a priori calculation of sample size, but we aimed for the same number of participants as Exp. 2. Participants were paid approximately 5 U$D in each session (20.000 COPs).

All procedures were in accordance with the ethical standards of the institutional research committee of the economics department of Universidad Javeriana that follows international

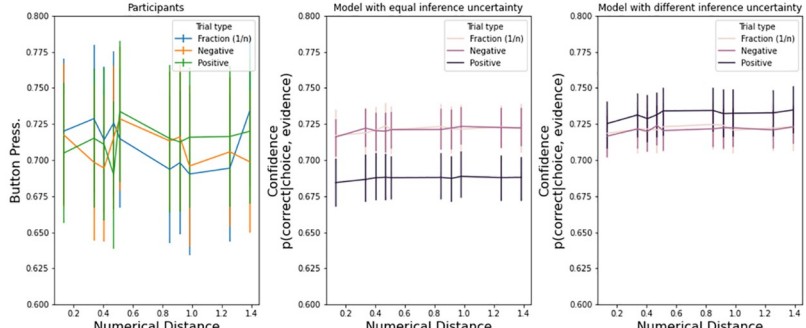

**Fig 9. Reduced uncertainty in trials with positive numerals.** Exp. 2: models with equal uncertainty had an UN = 4.5 for all types of numerals. Models with different uncertainty had $UN_{pos} = 3.5$, $UN_{neg} = 4.3$, $UN_{frac} = 4.3$.

and national norms regarding research with human subjects. The research committee approved the study with approval code: FCEA-DF-0433. All participants signed a written consent form after it was explained to them.

*Apparatus*, *stimuli*, *procedure*, *and design*. The apparatus, procedure, and design were like Exp. 2. The main differences were that in Exp. 3, non-dummy trials include all the single digits in the range 2 to 8, sampled randomly to form pairs. Also, digits were always blue, regardless of numeral type. Before starting, participants in Exp. 3 did 33 training trials where they received incorrect feedback (numbers turned red). Once they finished training, we turned off the red feedback in incorrect trials and they just saw blue digits for the 435 test trials regardless of performance. The objective of Exp. 3 was to eliminate any potential effect related to color cues or error feedback.

*Data analysis*. For Exp. 3, given that participants just did the numeral comparison task and there were no between-subjects variables such as experiments' order effects, we report fixed effects. They are like random effects: they control for subject level heterogeneity by adding a constant in the regression for each subject. But they do not make any independence assumptions between the constant and the other independent variables (random effects do make such assumption; in Exp. 1 and 2 we had to use random effects because fixed effect do not allow between-subject variables such as order of experiments).

## Results

Mean accuracy was almost at ceiling (Fig 10; Exp 3 (mean, std). 1/pos: 0.95, 0.06 neg: 0.93, 0.09 pos:0.93, 0.05). Table 3.1 reveals that accuracy was slightly higher with negative numerals and 1/n numerals. Accuracy improved with larger numerical distances.

In experiment 3, we also confirm the presence of confidence properties in button pressure and that numeral type modulated such pressure. We present three regressions, one for each panel of the confidence theory represented in Fig 1, while controlling for response times.

Accuracy and confidence should have a positive relation (Fig 1, left panel). Table 3.1 confirms this for Exp. 3. The pressure estimate is positive meaning that trials with higher pressure were more likely to be correct. Moreover, a simple regression, just including button pressure as a regressor, to directly test the theoretical confidence property in the first panel in Fig 1, also finds a positive relation (Exp. 3: $\beta = 0.07$, 95%CI = [0.05, 0.09], p <0.01).

For a given discriminability (i.e. numerical distance) trials with high confidence, as defined by a median split, should be more accurate (Fig 1, right panel). Table 3.2. reveals that a regressor for a dummy for the median split of max. button pressure is significant. This means that when subjects were correct, they pressed the force sensor harder, controlling for numerical distance. Moreover, a simple regression, just including the median split and numerical distance as regressors, to directly test the theoretical confidence property in the third panel in Fig 1, also finds a positive estimate for the median split (Exp. 3: $\beta = 0.02$, 95%CI = [0.01, 0.02], p <0.01).

Correct trials should have higher levels of confidence than incorrect trials for a given discriminability i.e. numerical distance (Fig 1, central panel). Table 3.3 shows that indeed correct trials have higher button pressure (correct estimate), in a regression that controls for numerical distance. Moreover, a simple regression, excluding numeral types and just including a dummy for correct trials and numerical distance, to directly test the theoretical confidence property in the central panel in Fig 1, further confirms this overall higher button pressure in correct trials (Exp. 3: $\beta = 0.07$, 95%CI = [0.05, 0.09], p<0.01). Fig 1, central panel, shows distinct slopes for correct and incorrect trials. However, the interaction term was not significant,

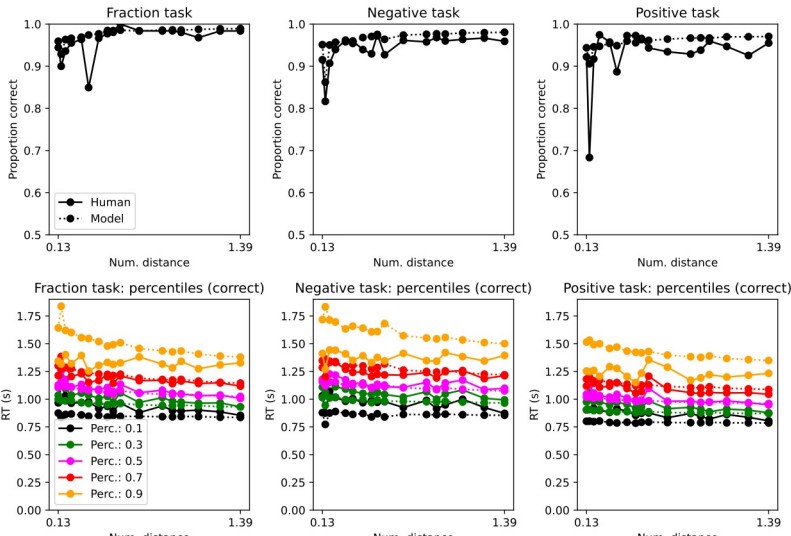

**Fig 10. Accuracy (top) and response time distributions by percentiles (bottom) by trial type (Exp 3.).** Numerical distance in log. scale. Response times are for correct trials; see Supplemental Information for incorrect trials.

but it was positive in line with the theory. Our participants had a low error rate and could explain the lack of interaction.

Fig 11 has a visualization of all the confidence properties in max. button pressure that we explained in the previous paragraphs. Accuracy was higher with stronger button pressures (left panel). Button pressure was higher in correct trials (center panel). For a given difficulty, accuracy was higher in trials with higher confidence (right panel). The presence of these three properties indicates that participants expressed their confidence level with button pressure.

Response times behaved similarly as in previous number cognition research (Exp. 3 (mean, std): 1/pos: 1096 ms, 129 ms; neg: 1154 ms, 125 ms; pos: 1030 ms, 123 ms). Participants were slower in inverted trials (negative and 1/n numerals). There was an effect of numerical distance. As the distance between numerals increased response times got faster (Table 3.4). In Exp. 3, the interaction terms were not significant but had a negative sign as in Exp. 1 and 2.

**Table 3.1. Accuracy.** Exp. 3. Fixed Effects Estimation.

|  | Par. | Std.Err | t | p | Low CI | High CI |
|---|---|---|---|---|---|---|
| Intercept | 0.95 | 0.01 | 80.91 | 0 | 0.93 | 0.98 |
| 1/n | 0.03 | 0 | 8.06 | 0 | 0.02 | 0.04 |
| Neg. | 0.02 | 0 | 3.96 | 0 | 0.01 | 0.03 |
| Max. press | 0.08 | 0.01 | 7.72 | 0 | 0.06 | 0.1 |
| Num. dist | 0.01 | 0 | 3.49 | 0 | 0.01 | 0.02 |
| RT | -0.07 | 0.01 | -7.45 | 0 | -0.09 | -0.05 |
| Cov. Estimator: | Robust | Log-likelihood | 3062.9 | F (5,15132): | 34.622 |  |
| No. subj: | 45 | No. Obs: | 15182 | P-value | 0 |  |
| BIC: | -6068 | BIC vs Linear: | -3 |  |  |  |

**Table 3.2. Accuracy. Exp. 3.** Fixed Effects Estimation.

| | Par. | Std.Err | t | p | Low CI | High CI |
|---|---|---|---|---|---|---|
| Intercept | 0.99 | 0.01 | 97.42 | 0 | 0.97 | 1.01 |
| 1/n | 0.03 | 0 | 8 | 0 | 0.02 | 0.04 |
| Neg. | 0.02 | 0 | 3.86 | 0 | 0.01 | 0.03 |
| Median max. press. (dummy) | 0.02 | 0 | 6.64 | 0 | 0.02 | 0.03 |
| Num. dist | 0.01 | 0 | 3.55 | 0 | 0.01 | 0.02 |
| RT | -0.07 | 0.01 | -7.17 | 0 | -0.08 | -0.05 |
| Cov. Estimator: | Robust | Log-likelihood | 3040.1 | F (5,15132): | 30.967 | |
| No. subj: | 45 | No. Obs: | 15182 | P-value | 0 | |
| BIC: | -6022 | BIC vs Linear: | -3 | | | |

## Drift diffusion results

As with Exp. 1 and 2., the drift-diffusion decision model captured mean accuracy and response time percentiles (Fig 10). Also, there was a change in decision dynamics across inverted (negative and 1/positive) and positive numerals (Table 1.3).

The model has the expected characteristics (Fig 12): a) accuracy improves with higher confidence, b) correct trials have large confidence, and c) trials with larger confidence, as determined by a median cut, are more accurate. The model is quantitatively different from the observed button pressure because we do not know (so we could not implement) a transfer function Confidence -> Button pressure.

As with Exp. 1 and 2, Fig 13 shows that if we assume equal uncertainty UN, trials with positive numbers produce less confidence (center panels). However, on the left panels, we present actual button pressure. Button pressure for positive numbers is not weaker, if anything stronger (Table 3.3). Given the observed accuracy and response times used to estimate the DDM parameters, negative numerals seem to induce higher uncertainty UN.

## Discussion

We measured confidence in symbolic single-digit comparisons with button pressure and a computational model. Both sources of information pointed to a reduced level of confidence in negative numerals. First, button pressure contained signatures of confidence and was weaker for negative numerals. Second, the drift diffusion model also suggested a higher uncertainty

**Table 3.3. Max. Press.** Exp 3. Fixed Effects Estimation.

| | Par. | Std.Err | t | p | Low CI | High CI |
|---|---|---|---|---|---|---|
| Intercept | 0.48 | 0.02 | 24.34 | 0 | 0.44 | 0.52 |
| 1/n | -0.01 | 0 | -2.25 | 0.02 | -0.02 | 0 |
| Neg. | -0.01 | 0 | -3.05 | 0 | -0.02 | 0 |
| Correct | 0.07 | 0.02 | 3.8 | 0 | 0.03 | 0.1 |
| Num. dist | 0 | 0.02 | -0.02 | 0.98 | -0.04 | 0.04 |
| Dist:Correct | 0.01 | 0.02 | 0.43 | 0.67 | -0.03 | 0.05 |
| RT | 0.1 | 0.01 | 13.42 | 0 | 0.08 | 0.11 |
| Cov. Estimator: | Robust | Log-likelihood | 3520.1 | F (6,15131): | 38.314 | |
| No. subj: | 45 | No. Obs: | 15182 | P-value | 0 | |
| BIC: | -6973 | BIC vs Linear: | -3 | | | |

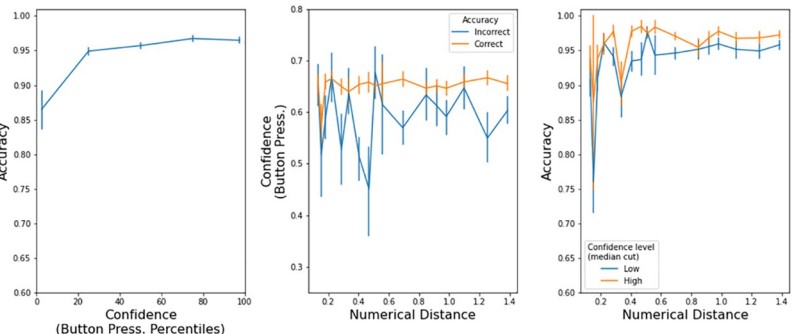

**Fig 11. Confidence characteristics in button pressure (Exp. 3).** Error bars are s.e.m.

parameter for negative and 1/positive numerals. Thus, in the model, negative numerals increase confidence slower than positive numerals (Eq 1). Third, decision dynamics were different for all types of numerals. For instance, trials with positive numerals seem to have higher symbolic facilitation. Such differences affect the decision variable, which affects confidence estimates (in our case, via Eq 1). We now turn to the general implications of these results.

## Reduced choice-confidence in negative numerals

We proposed button pressure as an implicit proxy for confidence. It had three theory-based characteristics of confidence and a positive correlation with response times; a correlation that was also present in a computational model that inferred the probability of being correct given the available evidence. Still, there are at least two alternatives: a) button pressure is only reflecting familiarity, or b) button pressure is only reflecting attention. That is, they are the sole drivers of button pressure so that confidence has nothing to do with our results.

The familiarity hypothesis predicts that more familiar objects, in our case positive numerals, induce faster response times [33]. Interestingly, we found a positive relation between response times and pressure. We could not find a theory on how familiarity translates to button pressure, but we think that a priori the prediction is that the more familiar object generates a stronger motor output. Thus, it seems to predict a negative relation (not a positive one): smaller response times for familiar objects should induce higher button pressure. Our model with confidence explains the positive relation: mores samples reduce uncertainty and improves confidence at choice (Eq 1).

**Table 3.4. RT. Exp. 3**. Fixed Effects Estimation Summary.

| | Par. | Std.Err | t | p | Low CI | High CI |
|---|---|---|---|---|---|---|
| Intercept | 1.08 | 0.01 | 162.84 | 0 | 1.06 | 1.09 |
| 1/n | 0.08 | 0.01 | 8.36 | 0 | 0.06 | 0.1 |
| Neg. | 0.13 | 0.01 | 13.59 | 0 | 0.11 | 0.15 |
| Num. Dist | -0.06 | 0.01 | -8.6 | 0 | -0.08 | -0.05 |
| 1/n:Dist | -0.01 | 0.01 | -1.16 | 0.25 | -0.03 | 0.01 |
| Neg:Dist | -0.01 | 0.01 | -0.77 | 0.44 | -0.03 | 0.01 |
| Cov. Estimator: | Robust | Log-likelihood | 1283.9 | F (5,15132): | 216.08 | |
| No. subj: | 45 | No. Obs: | 15182 | P-value | 0 | |
| BIC: | -2510 | BIC vs Linear: | -18 | | | |

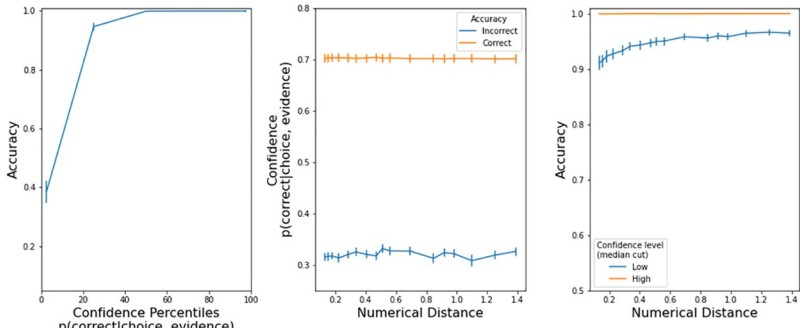

**Fig 12. Confidence properties in the drift diffusion model (Exp 3).** Left panels: accuracy increased with more confidence. Center panels: there was low confidence in incorrect trials. Right panels: accuracy is higher in high confidence trials. Error bars are s.e.m.

Attention, on the other hand, could produce the observed patterns of button pressure: a) more attention/pressure, more accuracy, b) more attention/pressure in correct trials, and c) higher accuracy in trials with higher attention/pressure (Fig 1). However, these three characteristics for confidence are based on a statistical theory [10] and such an accommodation of properties to attention would need theoretical validation demonstrating that they are solely about attention, not confidence. Moreover, there are no theories that negative numerals disengage attention and if our pressure results are solely about attention, it would still be an interesting empirical finding on its own.

Instead of thinking attention or familiarity as an exclusive explanation for our results, one possibility that we favor is to see these two alternatives as mechanisms supporting confidence. For instance, evidence improves faster with attended stimuli [34] and this affects confidence (via Eq 1).

We did not collect explicit self-reports (e.g., a Likert-type scale), as traditionally seen in confidence research. Thus, it remains unclear if similar effects appear with other measurements. Future work could further validate button pressure as a proxy for explicit confidence with correlational studies with self-reports. This would be interesting but would not invalidate the current results as self-reports and behavioral outcomes are not necessarily correlated, and they are usually weakly so [35].

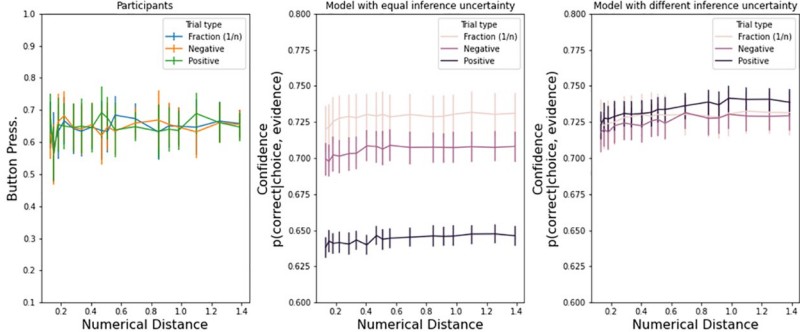

**Fig 13. Reduced uncertainty in trials with positive numerals.** Exp. 3 (bottom panels): models with equal uncertainty had an UN = 4.5 for all types of numerals. Models with different uncertainty had $UN_{pos} = 2.5$, $UN_{neg} = 4$, $UN_{frac} = 4.5$. Error bars are s.e.m.

Even in a simple task (single-digit comparisons), there was a detectable reduction in implicit confidence. Given the high accuracy in our participants (>90%), low confidence in abstract negative numbers may characterize their manipulation. There is a lingering trace of doubt when dealing with them. A higher uncertainty parameter UN in the model is a qualitative demonstration of a slower generation of information for negative numbers. This higher uncertainty could speak to the metacognition and learning of negatives and fractions. It is a well-established result in education literature that inverted numbers are hard for children and adults alike [36]. Also, there is an specific metacognition for mathematics [6]. The suggestion that negatives elicit less information is an interesting result that could link both lines of research. A question is if this higher uncertainty has direct links to quality of education, development trajectories or if better math abilities subdue some of the uncertainty.

There was a positive relationship between confidence and response time. The conditional probability p(correct|choice, evidence) could depend on response times via the number of samples (Eq 1). This dependency on the number of samples predicts that faster response times should be less confident. Interestingly, ours is not the first report with a positive relation [18]. This divide between studies that find positive and negative relations suggest distinct mechanisms to obtain confidence from internal decision variables. Here we proposed Eq 1; but confidence could also come from other decision-variable metrics, such as the standard deviation, that could correlate positively with response times.

## Negative number cognition: Holistic or strategic?

The DDM and response time results also speak to representational theories. Confidence was lower for negative numerals. Even though holistic and strategic theories predict such outcome, the underlying source is different. The holistic theory imputes the reduction of confidence to the holistic magnitudes for negative numerals. The strategic theory cannot impute changes in confidence to such holistic magnitudes for negative numerals because they do not exist: the mind only represents positive quantities.

Our results from button pressure alone cannot disentangle the hypotheses but we argue that reduced confidence cannot come only from strategic considerations. Prior theoretical work suggests that confidence reports come from decision variables [16]. The drift-diffusion model relied on the perceived numerical distance to produce a metric of confidence (Eq 1), and it replicated qualitative patterns of confidence. Also, the drift-diffusion dynamics changed across numeral types. This is also consistent with the possibility that negative numerals have distinct holistic magnitudes. For instance, the difference in drift rate for positive and negative numerals means that the decision variable in trials with positive and negative numerals was not comparable; even 1/positive trials had different decision-dynamics, consistent with the possibility of automatic activation of proportional magnitudes [37]. Still, this requires further research because the drift-diffusion model assumes independence between decision and non-decision times. If non-decision times permeate the decision variable in our task, then part of the effect could be imputable to non-decision features. But we argue that most likely both decision and non-decision times change between numeral types.

Response time results also favor holistic representations of negative numerals. The numerical distance slope of response times was different for numeral types in Exp. 1 and 2. Distinct response time slopes are an index of different sensitivity to numerical distance for negative numerals.

## Other theoretical insights for number cognition

We did not find an effect of numerical distance in button pressure (Table 3.3), in line with previous work that also report weak or no continuous force effect [13]. Still, we did find effects of numeral type, suggesting that number representations do affect motor planning and output. Our work was not designed to answer the underlying relationship between force and numerical distance. We designed the study and analyses to study confidence. Other designs, for instance, parity judgment tasks where the mental magnitude activates automatically, are better at addressing force-mental magnitude relationships.

We report that logarithmic distance was better at explaining the data than linear distance, as measured by lower BIC values. At face value, this means that our participants treated, for instance, 1 vs. 2 differently than 8 vs. 9 (and similarly with other comparable distances). In terms of the possible source of this effect, our experiment cannot clearly differentiate whether this is a consequence of having an internal logarithmic mental number line or a frequency effect such that some types of distances occur more often than others in the real world (e.g., via Benford's law). Both could also be happening as frequency effects relate to logarithm scales [38].

## Methodological aspects for number cognition

Another question is the applicability of the drift-diffusion framework to comparisons of negative numerals. The framework does not apply to multi-stage decisions [17]. The question then is if negative numbers are compared in a multi-stage fashion i.e. with many decision-variables. We argue it is simpler to assume that only one decision variable applies: numerical distance. Encoding, such as detecting the type of numeral on screen or dropping signs, is independent of the main decision loop that carries the decision-variable: perceived numerical distance. We assumed that non-decision times captured the detection of which type of numeral was on screen. Once detected the participant used the appropriate magnitudes. However, it is an assumption. Thus, our modeling results represent a particular context: when seeing mixed trials, negative numbers produce less information per unit of time (i.e., a larger UN parameter). That said, the overall confidence result was also present in the data. The model was a computational tool to gain further insights under the independence assumptions mentioned above.

The drift-diffusion model has been extensively applied in number cognition [27,28,39,40]. The framework has provided conceptual clarity regarding performance by allowing to integrate response times and accuracy into single measures, such as drift rate or thresholds. By modelling at the same time response time and accuracy it is possible to obtain a better characterization of behavior in number-related tasks. A better characterization is important because many studies report correlations between simple number tasks (e.g. ordering digits from smaller to larger) with high-level math abilities; while others do not [41,42]. Such empirical conflicts could be misinterpreted as failures to replicate but in fact could be a failure to integrate both accuracy and response times [27,40].

Concerning methods, we did not find a proper transfer function to model button pressure. In our literature search we did not find a fully developed model in the literature that specifies how cognitive representations of confidence interact with motor output leading to pressing a button. Step functions seem insufficient because button pressure reflects a continuous confidence signal (e.g. stronger for positive numbers), not easily explained by all-or-nothing button press.

The cognitive activation of negative magnitudes is context-dependent [8]. Therefore, in tasks where negative numerals comparisons are solved strategically (e.g., component model), confidence may not be affected. Our mixed design and the fact that participants did other

inversion-related tasks the same day could have helped activate holistic negative magnitudes. Though, we did not find spill-over effects in the regressions controlling for experiment order and Exp. 3, where participants only did one task, had similar outcomes. In sum, we provided evidence that negative magnitudes can carry less cognitive information than positive ones, even in a simple arithmetic task solved by educated adults. An intriguing question is if such a reduction of confidence translates to more complex tasks with negative quantities, and in which contexts (academic, economic, social). Negative numbers may produce more uncertainty, and this should impact learning, scientific reasoning, and decision-making.

## Supporting information

**S1 File.**
(DOCX)

## Author Contributions

**Conceptualization:** Santiago Alonso-Díaz, Gabriel I. Penagos-Londoño.

**Data curation:** Santiago Alonso-Díaz.

**Formal analysis:** Santiago Alonso-Díaz.

**Funding acquisition:** Santiago Alonso-Díaz.

**Investigation:** Santiago Alonso-Díaz, Gabriel I. Penagos-Londoño.

**Methodology:** Santiago Alonso-Díaz, Gabriel I. Penagos-Londoño.

**Project administration:** Santiago Alonso-Díaz, Gabriel I. Penagos-Londoño.

**Resources:** Santiago Alonso-Díaz, Gabriel I. Penagos-Londoño.

**Software:** Santiago Alonso-Díaz.

**Supervision:** Santiago Alonso-Díaz, Gabriel I. Penagos-Londoño.

**Validation:** Santiago Alonso-Díaz, Gabriel I. Penagos-Londoño.

**Visualization:** Santiago Alonso-Díaz.

**Writing – original draft:** Santiago Alonso-Díaz.

**Writing – review & editing:** Santiago Alonso-Díaz, Gabriel I. Penagos-Londoño.

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
