## [Decision Letter · Decision Letter 0]

31 Jan 2022

PONE-D-21-25521Reduced choice-confidence in negative numeralsPLOS ONE

Dear Dr. Alonso-Diaz,

Thank you for submitting your manuscript to PLOS ONE. After careful consideration, we feel that it has merit but does not fully meet PLOS ONE’s publication criteria as it currently stands. Therefore, we invite you to submit a revised version of the manuscript that addresses the points raised during the review process.

I received two reviews about your manuscript. Especially reviewer #1 raised some serious methodological concerns that need to be addressed. I encourage you to amend the manuscript according to the suggestions, taking special care of the points raised by reviewer #1. Note that this can imply additional experimental work, according to how the comment about a potential design flaw is addressed. When sublitting your revison, please double check figures and upload high quality artwork, current figures are barely readable. 

We look forward to receiving your revised manuscript.

Kind regards,

Federico Giove, PhD

Academic Editor

PLOS ONE

https://journals.plos.org/plosone/s/file?id=ba62/PLOSOne_formatting_sample_title_authors_affiliations.pdf”

“The study received funding from the university.”We note that you have provided funding information that is not currently declared in your Funding Statement. However, funding information should not appear in the Acknowledgments section or other areas of your manuscript. We will only publish funding information present in the Funding Statement section of the online submission form.

“S.A. received an early career grant from the university (Pontificia Universidad Javeriana. ID PPTA 8329; https://www.javeriana.edu.co/vicerrectoria-de-investigacion/). The funders had no role in study design, data collection and analysis, decision to publish, or preparation of the manuscript”

Reviewers' comments:

Reviewer's Responses to Questions

**Comments to the Author**

1. Is the manuscript technically sound, and do the data support the conclusions?

Reviewer #1: No

Reviewer #2: Yes

2. Has the statistical analysis been performed appropriately and rigorously? 

Reviewer #1: No

Reviewer #2: Yes

3. Have the authors made all data underlying the findings in their manuscript fully available?

Reviewer #1: No

Reviewer #2: Yes

4. Is the manuscript presented in an intelligible fashion and written in standard English?

Reviewer #1: No

Reviewer #2: Yes

5. Review Comments to the Author

Reviewer #1: Summary

The authors report statistical modeling of two data sets in which adults performed magnitude comparisons with positive, negative, and fraction numerals. Their main point is that the second data set revealed weaker maximum response force for negative compared to the other number types.

Evaluation

While the main message is newsworthy, the ms fails to integrate this finding adequately into the current literature and consequently lacks important methodological and analytical detail. Furthermore, a potential flaw in the design might account for the result and needs to be addressed. A substantial revision is necessary before this ms could perhaps make a useful contribution to the field.

Major Problems

1) LITERATURE: The authors seem unaware of the distinction between kinematics and kinetics of movement. While they cite a range of kinematic studies (mouse tracking, pointing), they completely overlooked the cognitive literature on force production, beginning with Abrams & Balota (1991) and extending to the recent modeling work of Miklashevsky et al. (2021) in the numerical domain. This omission results in rather superficial reporting for the “novel” dependent measure, both in terms of data collection and data analysis (see below). Another example is the authors’ referring to Dotan’s work for log compression (p. 21) but the same authors have since revoked this account (Dotan & Dehaene, 2016).

2) METHODS AND DESIGN: The information contained on p. 6-8 is incomplete and needs to be massively expanded and systematized (separate sections for participants, apparatus, stimuli, design, procedure) in order to allow proper appreciation and replication. A key point to elaborate is the recording and subsequent analysis of force data (see Miklashevsky et al., 2021 and references therein for the complexity of this topic) to help readers understand the choice and extraction of the specific force measure used. The relationship between number magnitude and force should also be reported to relate this work to the current debate. I list here several other specific omissions:

a) There is contradictory information about the range of numbers used (either 2-15 or 2,3,5,7,8; and why not 4 and 6?) and the specific items and their frequency that resulted in the reported number of trials.

b) What is a “dummy trial”?

c) What is meant by “anchor strategies” (illustrative example and references needed)?

d) The sample size is not justified, either a priori or retrospectively.

e) There seems to be no specific ethics approval for this study (as indicated by a reference number), merely a general statement that authors complied with ethical regulations.

f) The data collection was embedded into a series of related tasks, apparently intended to prime “inversion” (motion perception, categorization) that is not sufficiently well reported to permit understanding of possible spill-over effects; ideally, absence of spill-over should be formally reported in terms of non-significant order effects.

3) RESULTS

a) While there is extensive statistical modeling, some basics remain opaque because of lack of descriptives, such as reporting of average RT or accuracy in the text. One example is the differential distance effect in accuracy (p. 11) and speed; also, the authors confused “two samples” with “two-sided” testing (p. 10, bottom).

b) The Figures in the ms are of poor quality, making even the identification of axes labels impossible. This is unprofessional. All I was able to notice is that the authors erroneously used the unit “percent” for a probability scale.

4) DISCUSSION:

a) A potential flaw in the design might account for the result and needs to be addressed. Specifically, the decision to feed back errors through color changes (p.7) established specific color transition probabilities that would be effective at the start of each new trial. This may well be the very mechanisms by which “incorrect trials priors somehow moved” (p. 18) that the authors speculate about. Should this not be ruled out with a control experiment?

b) The authors never fulfill their promise (from p. 10) to discuss implications of the assumed seriality of processes (I found line 6 on p. 21 as the only return to this important issue)

Minor Issues: These largely relate to the writing style; in light of their sheer number it is worth highlighting them:

- The authors frequently state incomplete comparisons. Already the third sentence needs to be completed “…worse for negative numerals than for …”; there are numerous such instances throughout the ms.

- The ms contains a large number of grammatically problematic formulations. A pertinent example that misleads readers is on page 8, line 4 from the bottom, where the authors stated ”It includes...” but should have stated “They include…” because it is NOT the decision variable that includes encoding and responding-related times.

- The writing is often opaque because of colloquial style (e.g., first new paragraph on p. 19). But already the very first sentence “Negatives are essential in mathematics” requires elaboration to rule out photographic negatives.

- There are some illogical statements. For example, the sentences 5 and 6 of the Intro are contradictory because lack of variability prevents strong correlations. On page 9 various cognitive processes are ascribed to the minus B parameter.

- Further intransparency results from the use of different labels for the same concept (1/positive numerals, inverted numbers, inverted problems, later 1/n fractions) and lack of definition of acronyms such as BIC (p.9).

- The ms should be checked for typos (e.g., p. 9: “striping” should be “stripping”, p. 18: “between response times” should be “with response times”)

References

Abrams, R. A., & Balota, D. A. (1991). Mental chronometry: Beyond reaction time. Psychological Science, 2, 153-157.

Dotan, D., & Dehaene, S. (2016). On the origins of logarithmic number-to-position mapping. Psychological Review, 123(6), 637–666. doi:10.1037/rev0000038

Miklashevsky, A., Lindemann, O. & Fischer, M.H. (2021). The Force of Numbers: Investigating Manual Signatures of Embodied Number Processing. Front. Hum. Neurosci. 14:590508. doi: 10.3389/fnhum.2020.590508

Reviewer #2: This paper presents experiments and models on the confidence of decision (post-decision estimate of correct) in the case of negative numbers.

It is an important issue, as many evidences both from human children and animals suggest that their coding is different from small positive numerosities and numbers.

I find particularly interesting that authors complement the study with human participants and the model.

About this, I suggest to add a wider reference on models proposed in numerical cognition. More in general, as it is a very wide field, adding more references can be useful.

Results confirm authors' hypothesis and are interesting for a wide audience; I suggest to stress the contribution of the model in the discussion.

6. PLOS authors have the option to publish the peer review history of their article (what does this mean?). If published, this will include your full peer review and any attached files.

Reviewer #1: No

Reviewer #2: No

---

## [Author Response · Author response to Decision Letter 0]

3 Apr 2022

Reviewer #1: Summary

We want to start by thanking you for your detailed review, is something we truly appreciate, and hope to fully address your main concerns below and in the new manuscript. 

The authors report statistical modeling of two data sets in which adults performed magnitude comparisons with positive, negative, and fraction numerals. Their main point is that the second data set revealed weaker maximum response force for negative compared to the other number types.

Evaluation

While the main message is newsworthy, the ms fails to integrate this finding adequately into the current literature and consequently lacks important methodological and analytical detail. Furthermore, a potential flaw in the design might account for the result and needs to be addressed. A substantial revision is necessary before this ms could perhaps make a useful contribution to the field.

Major Problems

1) LITERATURE: The authors seem unaware of the distinction between kinematics and kinetics of movement. While they cite a range of kinematic studies (mouse tracking, pointing), they completely overlooked the cognitive literature on force production, beginning with Abrams & Balota (1991) and extending to the recent modeling work of Miklashevsky et al. (2021) in the numerical domain. This omission results in rather superficial reporting for the “novel” dependent measure, both in terms of data collection and data analysis (see below). Another example is the authors’ referring to Dotan’s work for log compression (p. 21) but the same authors have since revoked this account (Dotan & Dehaene, 2016).

• Indeed, we had a bias towards kinematic studies, which we are more familiar with; this is our first force paper. We are grateful for these new references relating force and number cognition. In the introduction we tried to summarize the overall findings of this literature. page 4

• We also drop any mention that our paper uses a “novel” dependent measure. We are now aware that it is not the case.

• Regarding the Dotan paper of 2013, we no longer use it in the new manuscript.

2) METHODS AND DESIGN: The information contained on p. 6-8 is incomplete and needs to be massively expanded and systematized (separate sections for participants, apparatus, stimuli, design, procedure) in order to allow proper appreciation and replication. A key point to elaborate is the recording and subsequent analysis of force data (see Miklashevsky et al., 2021 and references therein for the complexity of this topic) to help readers understand the choice and extraction of the specific force measure used. 

• We extend the explanation of the apparatus and measure (page 7) and provide a diagram of the Arduino setup in supplemental information (Supp. Fig. 13).

The relationship between number magnitude and force should also be reported to relate this work to the current debate. 

• Tables 3 report this relationship. It was not significant. We did find lower button pressure for negative numerals than for positive numerals, suggesting that the nature of the number representations do affect motor planning. We discuss this in page 27.

I list here several other specific omissions:

a) There is contradictory information about the range of numbers used (either 2-15 or 2,3,5,7,8; and why not 4 and 6?) and the specific items and their frequency that resulted in the reported number of trials.

• We put a table with the actual numbers and frequencies (page 9). 

• We improved the explanation on why the set 2,3,5,7, and 8, and the explanation on why we distinguished between dummy (2-15) and non-dummy trials (2,3,5,7,and, 8) (pages 7-9) 

• We ran a new experiment with the full set 2,3,4,5,6,7,8.

b) What is a “dummy trial”?

• A trial that exposes participants to a wider range of numbers but, and this is important, we do not include in data analysis. We clarify in the new manuscript why these trials are important to avoid anchor strategies (which we also explain better in the new manuscript and see our response to c) below). Pages 7-9

c) What is meant by “anchor strategies” (illustrative example and references needed)?

• In the set of numbers 1 to 9, 1 and 9 are the extreme anchors. Thus, if participants experience that set of numbers, every time they see a 1 or 9 in the pair they don’t have to compute any distance, they just pick the other number (if 1) or 9.

• Thus, dummy trials include 9 (the range of dummy trials was 2 to 15) but we do not analyze them (also because in dummy trials there are two-digit numbers).

• Including dummy trials was a conservative approach to avoid strategic behavior so that participants mostly based their judgments on numerical distance.

d) The sample size is not justified, either a priori or retrospectively.

• We now further clarify in the methods a retrospective reason (we preregistered 50), and give a rationale why we do not calculate post-hoc power. Page 6

• Sample size for the new experiment was capped at 50.

e) There seems to be no specific ethics approval for this study (as indicated by a reference number), merely a general statement that authors complied with ethical regulations.

• Sorry if this was not clear. We now further emphasize that the study was approved by an actual committee and the reference number. Page 7

f) The data collection was embedded into a series of related tasks, apparently intended to prime “inversion” (motion perception, categorization) that is not sufficiently well reported to permit understanding of possible spill-over effects; ideally, absence of spill-over should be formally reported in terms of non-significant order effects.

• We change all the regressions to random effects panel regressions to include the between-subjects factor of order (page 9 and 10). Fixed-effects regressions by definition do not accept between-subject variables. The overall findings that numeral type affected button pressure holds, even after controlling for order. Moreover, the presence of theoretically relevant effects of confidence in button pressure do not disappear after controlling for order (Tables 2,3,4,5)

• We ran a new experiment where participants only did the numeral comparison task and the results were similar.

3) RESULTS

a) While there is extensive statistical modeling, some basics remain opaque because of lack of descriptives, such as reporting of average RT or accuracy in the text. One example is the differential distance effect in accuracy (p. 11) and speed; 

• We now include average and standard deviations for accuracy (page 14) and response time (page 18), by numeral type.

• Also, Figure 4 has the average accuracy and response times by percentiles, both for participants and the DDM model.

also, the authors confused “two samples” with “two-sided” testing (p. 10, bottom).

• We did mean two-sample t-test (i.e. a paired independent-samples t test). We clarify this in the new manuscript (Table 5, pages 20 and 21).

b) The Figures in the ms are of poor quality, making even the identification of axes labels impossible. This is unprofessional. All I was able to notice is that the authors erroneously used the unit “percent” for a probability scale.

• We are sorry for this. We submitted high quality figures however it seems that the ones rendered by the submission system on the manuscript were low quality ones. 

• The high-quality ones had to be accessed via the link in the upper left corner of the page containing each figure.

• In case the rendering system fails, we now uploaded the manuscript with figures in place to psyarxiv: https://psyarxiv.com/sdfb9

4) DISCUSSION:

a) A potential flaw in the design might account for the result and needs to be addressed. Specifically, the decision to feed back errors through color changes (p.7) established specific color transition probabilities that would be effective at the start of each new trial. This may well be the very mechanisms by which “incorrect trials priors somehow moved” (p. 18) that the authors speculate about. Should this not be ruled out with a control experiment?

• We now include in the regressions a dummy variable controlling if the current trial was preceded by an incorrect response. The overall results were the same, which discards a simple color transition effect (Tables 2,3,4).

• Moreover, there are really few incorrect trials (approx. 5%), so such transitions were experienced rarely in most participants.

• Finally, we also ran a new experiment without feedback or any color changes and we obtained similar results

b) The authors never fulfill their promise (from p. 10) to discuss implications of the assumed seriality of processes (I found line 6 on p. 21 as the only return to this important issue)

• We agree that this is an important issue but we do not have the means to confirm the seriality. We apologize if in the original manuscript there was any indication that our paper was about seriality. For this revision, we drop any promise and discussion, and just state in the introduction that it is an important assumption of the drift diffusion framework. We think is important to make this assumption transparent for future researchers using the DDM model.

Minor Issues: These largely relate to the writing style; in light of their sheer number it is worth highlighting them:

- The authors frequently state incomplete comparisons. Already the third sentence needs to be completed “…worse for negative numerals than for …”; there are numerous such instances throughout the ms.

• Hopefully we completed all comparisons (we apologize if we still have any in the new manuscript)

- The ms contains a large number of grammatically problematic formulations. A pertinent example that misleads readers is on page 8, line 4 from the bottom, where the authors stated ”It includes...” but should have stated “They include…” because it is NOT the decision variable that includes encoding and responding-related times.

• We fixed this and (hopefully) similar ones.

- The writing is often opaque because of colloquial style (e.g., first new paragraph on p. 19). But already the very first sentence “Negatives are essential in mathematics” requires elaboration to rule out photographic negatives.

• Sorry for this. We did not try to be opaque and agree that science should be clear. We had colleagues read the original manuscript but it was not enough. For this new submission we tried to revise any opaque writing. Also, even though we know is just but an example you brought, we changed the expression “negatives” with the more specific “negative numerals” or “negative numbers”.

- There are some illogical statements. For example, the sentences 5 and 6 of the Intro are contradictory because lack of variability prevents strong correlations. 

• We erased those sentences. We wanted to say that high accuracy relates to high confidence, and this is a usual finding. However, we decided to drop that idea as it is not critical for the argument.

On page 9 various cognitive processes are ascribed to the minus B parameter.

• We are not sure if we understand your point here. B is compression and minus is the rotation to the negative portion. We now further clarify this in the new manuscript.

- Further intransparency results from the use of different labels for the same concept (1/positive numerals, inverted numbers, inverted problems, later 1/n fractions) and lack of definition of acronyms such as BIC (p.9).

• We changed all 1/n (or similar) to 1/positive numerals.

• We now define the acronym BIC.

- The ms should be checked for typos (e.g., p. 9: “striping” should be “stripping”, p. 18: “between response times” should be “with response times”)

• Done (hopefully).

References

Abrams, R. A., & Balota, D. A. (1991). Mental chronometry: Beyond reaction time. Psychological Science, 2, 153-157.

Dotan, D., & Dehaene, S. (2016). On the origins of logarithmic number-to-position mapping. Psychological Review, 123(6), 637–666. doi:10.1037/rev0000038

Miklashevsky, A., Lindemann, O. & Fischer, M.H. (2021). The Force of Numbers: Investigating Manual Signatures of Embodied Number Processing. Front. Hum. Neurosci. 14:590508. doi: 10.3389/fnhum.2020.590508

Reviewer #2: This paper presents experiments and models on the confidence of decision (post-decision estimate of correct) in the case of negative numbers.

It is an important issue, as many evidences both from human children and animals suggest that their coding is different from small positive numerosities and numbers.

I find particularly interesting that authors complement the study with human participants and the model.

About this, I suggest to add a wider reference on models proposed in numerical cognition. More in general, as it is a very wide field, adding more references can be useful.

Results confirm authors' hypothesis and are interesting for a wide audience; I suggest to stress the contribution of the model in the discussion.

• Thanks for your comments and noting the significance of the paper. We appreciate it. 

• We now stress the contribution of the model. In particular, in regard to confidence Pages 27-28.

• We do not compare our model to other non-ddm models (e.g. Huber, et al, 2016 connectionist model) because it will be too hard and speculative. 

---

## [Decision Letter · Decision Letter 1]

22 Apr 2022

PONE-D-21-25521R1Reduced choice-confidence in negative numeralsPLOS ONE

Dear Dr. Alonso-Diaz,

Thank you for submitting your manuscript to PLOS ONE. After careful consideration, we feel that it has merit but does not fully meet PLOS ONE’s publication criteria as it currently stands. Therefore, we invite you to submit a revised version of the manuscript that addresses the points raised during the review process. There are still many comments of reviewer #1 that should be addressed, in particular all those that are related to methodological issues or unclear procedures. Considering the split reviewers opinions, I'm involving a third reviewer. Note that the third reviewer will receive the current or the revised version of your paper according to the time needed for revision submission. In any case, please add to your next revision a detailed response to each point raised by reviewer #1.

We look forward to receiving your revised manuscript.

Kind regards,

Federico Giove, PhD

Academic Editor

PLOS ONE

Reviewers' comments:

Reviewer's Responses to Questions

**Comments to the Author**

1. If the authors have adequately addressed your comments raised in a previous round of review and you feel that this manuscript is now acceptable for publication, you may indicate that here to bypass the “Comments to the Author” section, enter your conflict of interest statement in the “Confidential to Editor” section, and submit your "Accept" recommendation.

Reviewer #1: (No Response)

Reviewer #2: All comments have been addressed

2. Is the manuscript technically sound, and do the data support the conclusions?

Reviewer #1: Partly

Reviewer #2: Yes

3. Has the statistical analysis been performed appropriately and rigorously? 

Reviewer #1: I Don't Know

Reviewer #2: Yes

4. Have the authors made all data underlying the findings in their manuscript fully available?

Reviewer #1: Yes

Reviewer #2: Yes

5. Is the manuscript presented in an intelligible fashion and written in standard English?

Reviewer #1: No

Reviewer #2: Yes

6. Review Comments to the Author

Reviewer #1: I have read the cover letter and the revised ms plus the supplementary document and my impression is mixed.

1. The authors made several adjustments and additions that strengthen the ms, including explanations of their idiosyncratic terminology (“dummy trials” for filler trials) and references to relevant papers that were previously omitted. However, repeatedly directing readers to Ganor-Stern & Tzelgov (2008) for the component model is still suboptimal because the paper by Huber (their Ref 5) constitutes its more recent development. Similarly, the introduction of “button pressure” as the measure of interest (on page 4) is immediately followed by references to “motor metrics”, i.e. kinematic studies, even though the “button pressure” is an isometric assessment without any kinematics involved. In my idiosyncratic view these are suboptimal revisions.

2. More importantly, the methods descriptions are still not detailed enough to permit replication. Although stimulus ranges are now clear and the recording apparatus for “button pressure” is now explained in more detail, the sensitivity of this device is still unclear. If the range is 0.25 to 0.90, does this mean that 65 levels of pressure were discriminable while the actual force produced (in Pascal or Newton) remains unknown? Another open issue, of fundamental importance for the later discussion about early attentional vs later conceptual effects: What was the sampling frequency (i.e. the temporal resolution of pressure measurements)?

3. Moreover, despite my extensive queries on this point, I still find only a SINGLE sentence that describes the force data analysis, namely “Data analysis. We used panel linear regressions to analyze the accuracy (linear probability model), response times up to two standard deviations from the mean (i.e., 95% of the trials), and button pressure”. This leaves open fundamental questions such as the time during which force was recorded or integrated, the data filtering or trimming for this inherently noisy signal, or the computation of parameters for analysis, such as average or peak force per trial, or many other candidates. None of this is explained.

4. I am unhappy with various minor aspects of this revision: The authors’ claim to defend their sample size of 50 participants “in studies with similar sample sizes (1,2,4,8)” (p. 6) is misleading readers because only Experiment 2 of ref. 4 has 55 participants, while all others have between 16 and 27. My request to determine power or sensitivity retrospectively was not addressed. Some wording is still poor: On page 7 “Exp. 1 was run in a 13-inch laptop” (should be “on”), or “Uncertainty is a more general concept as it is not a conditioned in choice.” (p. 5) is ungrammatical. Several references contain the superfluous word “internet”.

Reviewer #2: The authors have addressed most of the raised points. In my previous revision I focused on the model, as it is interesting in my opinion and authors have included a wider reflection on this issue

7. PLOS authors have the option to publish the peer review history of their article (what does this mean?). If published, this will include your full peer review and any attached files.

Reviewer #1: No

Reviewer #2: No

---

## [Author Response · Author response to Decision Letter 1]

24 May 2022

PLEASE SEE OUR RESPONSES ON THE WORD FILE AS THERE ARE SOME FIGURES

Reviewer #1: 

I have read the cover letter and the revised ms plus the supplementary document and my impression is mixed.

1. The authors made several adjustments and additions that strengthen the ms, including explanations of their idiosyncratic terminology (“dummy trials” for filler trials) and references to relevant papers that were previously omitted. However, repeatedly directing readers to Ganor-Stern & Tzelgov (2008) for the component model is still suboptimal because the paper by Huber (their Ref 5) constitutes its more recent development. 

We now extend the component model explanation with the Huber et al neural network (Intro.: Pages 3-4). 

Similarly, the introduction of “button pressure” as the measure of interest (on page 4) is immediately followed by references to “motor metrics”, i.e. kinematic studies, even though the “button pressure” is an isometric assessment without any kinematics involved. In my idiosyncratic view these are suboptimal revisions.

We wanted to show to a general audience that “motor metrics” are used in cognition. We understand that those references were about kinematics. Thus, we now drop those kinematics references to avoid any confusion. 

2. More importantly, the methods descriptions are still not detailed enough to permit replication. Although stimulus ranges are now clear and the recording apparatus for “button pressure” is now explained in more detail, the sensitivity of this device is still unclear. If the range is 0.25 to 0.90, does this mean that 65 levels of pressure were discriminable 

The range 0.25-0.90 is continuous. There are more than 65 discrete levels. 

The actual range from the raw data is 0.025 to 1 (Exp 2) and 0.05 to 1 (Exp. 3). The 0.25 – 0.9 range was after taking averages per numerical distances. In the new manuscript (Data analysis: Page 10) we decided that it was clearer if we present the ranges from the raw data because the right end is 1 and the left range is close to zero.

while the actual force produced (in Pascal or Newton) remains unknown? 

The sensor detects pressure as a change in resistance. The details of the resistors and Arduino wiring are in the manuscript and supplemental information. The Arduino transforms those changes of resistance into a discrete signal between 0 and 1023. All subjects used the same Arduino setup (including voltage and resistors). Thus, we do not report Pascals or Newtons (the transformation from change in resistance to Newtons or Pascals is not trivial). Still, the change in resistance indeed relates to force; resistance-based pressure is used in many current devices including mobile phones and many portable devices.

Another open issue, of fundamental importance for the later discussion about early attentional vs later conceptual effects: What was the sampling frequency (i.e. the temporal resolution of pressure measurements)?

We now report the sampling frequency of the Arduino board in the manuscript (Apparatus, page 7-8) (approx. 26 hz but see manuscript for details). 

It is not fast enough to fully disentangle early attention and later concepts (holistic magnitudes). 

In the discussion we accept that it could be attention, but it would require some ad-hoc accommodations so that all our tables and results are about attention and not confidence. 

As for concepts, we do think that confidence results speak about concepts, namely the existence of negative holistic magnitudes (see discussion section). However, we are also aware that our paper only brings partial clarity to negative magnitude processing. Our paper is just about confidence.

3. Moreover, despite my extensive queries on this point, I still find only a SINGLE sentence that describes the force data analysis, namely “Data analysis. We used panel linear regressions to analyze the accuracy (linear probability model), response times up to two standard deviations from the mean (i.e., 95% of the trials), and button pressure”. 

We apologize for our lack of clarity; it is not our intention. In the first revision we tried to address all your comments, but we clearly fail. We hope that our responses below clarify our data analysis.

This leaves open fundamental questions such as the time during which force was recorded or integrated, 

We analyzed signals as soon as the numerals appeared on screen and until the subject selected an option and was no longer pressing the force sensor (i.e. force resistance < low_pressure_threshold) (this for all trials). (Data Analysis: Page 10).

the data filtering or trimming for this inherently noisy signal, 

We did not filter Arduino signals. When the sensor is pressed it sends a reading to Matlab based on the participant’s pressure. In the figure below, each trial has a stereotypical look: a peak. This means that participants had a brief contact with the sensor (here we plot the raw Arduino signal, before any standardization) (see all raw data in Supplemental Figures 18 and 19).

We filtered data on the behavior side. We reported in the manuscript two criteria: response time greater than 2 standard deviations from the mean of all the data and accuracy less than 85%. 

We did detect an anomaly thanks to your comment. We plotted all the raw pressure data and we detected one subject who used the keyboard instead of the Arduino sensors. The pressures of subj. 33443 (Exp. 2) were always at baseline levels (Supp. Figure 18). This subj did respond accurately to the task, meaning that they used the keyboard (we explain this on Page 6 of the new manuscript). 

The reason this could happen was that we coded the first experiment (Exp. 1) so that subjects had to use the keyboard. In Exp. 2 we used the same base code (plus the Arduino stuff) and even though we verbally explained to all subjects to use the Arduino sensors, we kept, by mistake, the original instructions of Exp. 1 on screen. Subj. 33443 was the only subject (across Exp. 2 and 3) who mistakenly used the keyboard (see Supplemental Figure 18). We excluded this subject from this revision. Importantly, the main pattern of results did not change. 

or the computation of parameters for analysis, such as average or peak force per trial, or many other candidates. None of this is explained.

For this revision we now provide a concrete example for further clarity (Data Analysis: Page 10). Specifically, we used for analysis peak force during the trial, normalized to the max peak force during the whole session as a normalizing constant (separately for the left and right sensor). For instance, if subject X max peak force to the left during his whole session was 984, then all left pressures during the session were divided by 984. We used this metric in the tables. That is, the regression parameters were obtained with standardized peak pressure. We now put explicitly in the regressions the name of the variable followed by “normed”.

4. I am unhappy with various minor aspects of this revision: The authors’ claim to defend their sample size of 50 participants “in studies with similar sample sizes (1,2,4,8)” (p. 6) is misleading readers because only Experiment 2 of ref. 4 has 55 participants, while all others have between 16 and 27. 

We dropped the expression “similar sample size” to the actual range of 16 to 55. Thus, our sample size is on the larger end.

My request to determine power or sensitivity retrospectively was not addressed.

We calculated power, using response times from our participants as it is known in the literature that confidence and response times correlate (Participants: Page 7). Overall, the obtained power is large.

Some wording is still poor: On page 7 “Exp. 1 was run in a 13-inch laptop” (should be “on”),

Solved.

 or “Uncertainty is a more general concept as it is not a conditioned in choice.” (p. 5) is ungrammatical. 

We dropped that expression and rather clarify the distinction of uncertainty and confidence in the model because it is more concrete for the reader: DDM model: Page 13: “It is important to highlight that uncertainty UN is not the same as confidence in the model. UN is the base standard deviation of the inference, while confidence is the cumulative probability that μdv is greater than zero after a choice is made (i.e. after dv arrives at one of the thresholds).” 

Several references contain the superfluous word “internet”.

We did not realize that Mendeley was doing that, sorry.

OTHER IMPORTANT CLARIFICATIONS 

• In Exp. 3, by mistake, we standardized to a value that was not the peak pressure in general but the initial peak value in the Arduino signal greater than a low threshold. We corrected this oversight and that is why some regression estimates of Exp. 3 are different in this current revision. Importantly the overall pattern/direction/significance of the effects is identical, just slightly different estimates (i.e. both constants are correlated because each trial is a time series). This oversight did not affect the conclusions and the corrections, if any, consolidated the overall results.

The oversight happened because in Exp. 3 we decided to change the serial function used by Matlab to communicate with the Arduino (it will be deprecated; the new one is serialport). When we changed the code we marked events slightly differently. During data analyses we missed changing some of the marks and we ended up using, mistakenly, the peak pressure across trials of the first pressure signal sent by the Arduino greater than a low-pressure threshold on each trial. As we mentioned, we corrected this in the current revision.

• In the Table 1 of the previous revision we didn’t include the variable controlling for error feedback. In this revision we fixed that.

• The only tables that include interaction effects are the ones theoretically relevant. In Page 11: “We included interactions in the regressions when it was theoretically relevant. Namely when the dependent variable was response times (distinct slopes by numeral type suggest different mental magnitudes) and pressure (Figure 1, central panel shows distinct slopes for correct and incorrect trials).”

Reviewer #2: 

The authors have addressed most of the raised points. In my previous revision I focused on the model, as it is interesting in my opinion and authors have included a wider reflection on this issue

---

## [Editor Report · Decision Letter 2]

30 May 2022

PONE-D-21-25521R2

Reduced choice-confidence in negative numerals

PLOS ONE

Dear Dr. Alonso-Diaz,

This is a formal decision, to allow authors to address wrong submission

Thank you for submitting your manuscript to PLOS ONE. After careful consideration, we feel that it has merit but does not fully meet PLOS ONE’s publication criteria as it currently stands. Therefore, we invite you to submit a revised version of the manuscript that addresses the points raised during the review process.

We look forward to receiving your revised manuscript.

Kind regards,

Federico Giove, PhD

Academic Editor

PLOS ONE
---

## [Author Response · Author response to Decision Letter 2]

31 May 2022

Reviewer #1: 

I have read the cover letter and the revised ms plus the supplementary document and my impression is mixed.

1. The authors made several adjustments and additions that strengthen the ms, including explanations of their idiosyncratic terminology (“dummy trials” for filler trials) and references to relevant papers that were previously omitted. However, repeatedly directing readers to Ganor-Stern & Tzelgov (2008) for the component model is still suboptimal because the paper by Huber (their Ref 5) constitutes its more recent development. 

We now extend the component model explanation with the Huber et al neural network (Intro.: Pages 3-4). 

Similarly, the introduction of “button pressure” as the measure of interest (on page 4) is immediately followed by references to “motor metrics”, i.e. kinematic studies, even though the “button pressure” is an isometric assessment without any kinematics involved. In my idiosyncratic view these are suboptimal revisions.

We wanted to show to a general audience that “motor metrics” are used in cognition. We understand that those references were about kinematics. Thus, we now drop those kinematics references to avoid any confusion. 

2. More importantly, the methods descriptions are still not detailed enough to permit replication. Although stimulus ranges are now clear and the recording apparatus for “button pressure” is now explained in more detail, the sensitivity of this device is still unclear. If the range is 0.25 to 0.90, does this mean that 65 levels of pressure were discriminable 

The range 0.25-0.90 is continuous. There are more than 65 discrete levels. 

The actual range from the raw data is 0.025 to 1 (Exp 2) and 0.05 to 1 (Exp. 3). The 0.25 – 0.9 range was after taking averages per numerical distances. In the new manuscript (Data analysis: Page 10) we decided that it was clearer if we present the ranges from the raw data because the right end is 1 and the left range is close to zero.

while the actual force produced (in Pascal or Newton) remains unknown? 

The sensor detects pressure as a change in resistance. The details of the resistors and Arduino wiring are in the manuscript and supplemental information. The Arduino transforms those changes of resistance into a discrete signal between 0 and 1023. All subjects used the same Arduino setup (including voltage and resistors). Thus, we do not report Pascals or Newtons (the transformation from change in resistance to Newtons or Pascals is not trivial). Still, the change in resistance indeed relates to force; resistance-based pressure is used in many current devices including mobile phones and many portable devices.

Another open issue, of fundamental importance for the later discussion about early attentional vs later conceptual effects: What was the sampling frequency (i.e. the temporal resolution of pressure measurements)?

We now report the sampling frequency of the Arduino board in the manuscript (Apparatus, page 7-8) (approx. 26 hz but see manuscript for details). 

It is not fast enough to fully disentangle early attention and later concepts (holistic magnitudes). 

In the discussion we accept that it could be attention, but it would require some ad-hoc accommodations so that all our tables and results are about attention and not confidence. 

As for concepts, we do think that confidence results speak about concepts, namely the existence of negative holistic magnitudes (see discussion section). However, we are also aware that our paper only brings partial clarity to negative magnitude processing. Our paper is just about confidence.

3. Moreover, despite my extensive queries on this point, I still find only a SINGLE sentence that describes the force data analysis, namely “Data analysis. We used panel linear regressions to analyze the accuracy (linear probability model), response times up to two standard deviations from the mean (i.e., 95% of the trials), and button pressure”. 

We apologize for our lack of clarity; it is not our intention. In the first revision we tried to address all your comments, but we clearly fail. We hope that our responses below clarify our data analysis.

This leaves open fundamental questions such as the time during which force was recorded or integrated, 

We analyzed signals as soon as the numerals appeared on screen and until the subject selected an option and was no longer pressing the force sensor (i.e. force resistance < low_pressure_threshold) (this for all trials). (Data Analysis: Page 10).

the data filtering or trimming for this inherently noisy signal, 

We did not filter Arduino signals. When the sensor is pressed it sends a reading to Matlab based on the participant’s pressure. In the figure below, each trial has a stereotypical look: a peak. This means that participants had a brief contact with the sensor (here we plot the raw Arduino signal, before any standardization) (see all raw data in Supplemental Figures 18 and 19).

We filtered data on the behavior side. We reported in the manuscript two criteria: response time greater than 2 standard deviations from the mean of all the data and accuracy less than 85%. 

We did detect an anomaly thanks to your comment. We plotted all the raw pressure data and we detected one subject who used the keyboard instead of the Arduino sensors. The pressures of subj. 33443 (Exp. 2) were always at baseline levels (Supp. Figure 18). This subj did respond accurately to the task, meaning that they used the keyboard (we explain this on Page 6 of the new manuscript). 

The reason this could happen was that we coded the first experiment (Exp. 1) so that subjects had to use the keyboard. In Exp. 2 we used the same base code (plus the Arduino stuff) and even though we verbally explained to all subjects to use the Arduino sensors, we kept, by mistake, the original instructions of Exp. 1 on screen. Subj. 33443 was the only subject (across Exp. 2 and 3) who mistakenly used the keyboard (see Supplemental Figure 18). We excluded this subject from this revision. Importantly, the main pattern of results did not change. 

or the computation of parameters for analysis, such as average or peak force per trial, or many other candidates. None of this is explained.

For this revision we now provide a concrete example for further clarity (Data Analysis: Page 10). Specifically, we used for analysis peak force during the trial, normalized to the max peak force during the whole session as a normalizing constant (separately for the left and right sensor). For instance, if subject X max peak force to the left during his whole session was 984, then all left pressures during the session were divided by 984. We used this metric in the tables. That is, the regression parameters were obtained with standardized peak pressure. We now put explicitly in the regressions the name of the variable followed by “normed”.

4. I am unhappy with various minor aspects of this revision: The authors’ claim to defend their sample size of 50 participants “in studies with similar sample sizes (1,2,4,8)” (p. 6) is misleading readers because only Experiment 2 of ref. 4 has 55 participants, while all others have between 16 and 27. 

We dropped the expression “similar sample size” to the actual range of 16 to 55. Thus, our sample size is on the larger end.

My request to determine power or sensitivity retrospectively was not addressed.

We calculated power, using response times from our participants as it is known in the literature that confidence and response times correlate (Participants: Page 7). Overall, the obtained power is large.

Some wording is still poor: On page 7 “Exp. 1 was run in a 13-inch laptop” (should be “on”),

Solved.

 or “Uncertainty is a more general concept as it is not a conditioned in choice.” (p. 5) is ungrammatical. 

We dropped that expression and rather clarify the distinction of uncertainty and confidence in the model because it is more concrete for the reader: DDM model: Page 13: “It is important to highlight that uncertainty UN is not the same as confidence in the model. UN is the base standard deviation of the inference, while confidence is the cumulative probability that μdv is greater than zero after a choice is made (i.e. after dv arrives at one of the thresholds).” 

Several references contain the superfluous word “internet”.

We did not realize that Mendeley was doing that, sorry.

OTHER IMPORTANT CLARIFICATIONS 

• In Exp. 3, by mistake, we standardized to a value that was not the peak pressure in general but the initial peak value in the Arduino signal greater than a low threshold. We corrected this oversight and that is why some regression estimates of Exp. 3 are different in this current revision. Importantly the overall pattern/direction/significance of the effects is identical, just slightly different estimates (i.e. both constants are correlated because each trial is a time series). This oversight did not affect the conclusions and the corrections, if any, consolidated the overall results.

The oversight happened because in Exp. 3 we decided to change the serial function used by Matlab to communicate with the Arduino (it will be deprecated; the new one is serialport). When we changed the code we marked events slightly differently. During data analyses we missed changing some of the marks and we ended up using, mistakenly, the peak pressure across trials of the first pressure signal sent by the Arduino greater than a low-pressure threshold on each trial. As we mentioned, we corrected this in the current revision.

• In the Table 1 of the previous revision we didn’t include the variable controlling for error feedback. In this revision we fixed that.

• The only tables that include interaction effects are the ones theoretically relevant. In Page 11: “We included interactions in the regressions when it was theoretically relevant. Namely when the dependent variable was response times (distinct slopes by numeral type suggest different mental magnitudes) and pressure (Figure 1, central panel shows distinct slopes for correct and incorrect trials).”

• The manuscript is on psyarxiv with figures on the manuscript (https://psyarxiv.com/sdfb9/)

Reviewer #2: 

The authors have addressed most of the raised points. In my previous revision I focused on the model, as it is interesting in my opinion and authors have included a wider reflection on this issue

---

## [Decision Letter · Decision Letter 3]

29 Jun 2022

PONE-D-21-25521R3Reduced choice-confidence in negative numeralsPLOS ONE

Dear Dr. Alonso-Diaz,

Thank you for submitting your manuscript to PLOS ONE. After careful consideration, we feel that it has merit but does not fully meet PLOS ONE’s publication criteria as it currently stands. Therefore, we invite you to submit a revised version of the manuscript that addresses the points raised during the review process.

One of the original reviewers (#1) suggested rejection, based on unsatisfactorily response to his/her comments. I involved a third reviewer. While generally praising your work, reviewer #3 raised a number of further concerns.I encourage the authors to address all the new criticism raised by reviewer #3, as well as trying to address the  residual part of comments of #1. Please include a response to reviewer covering also #1.

We look forward to receiving your revised manuscript.

Kind regards,

Federico Giove, PhD

Academic Editor

PLOS ONE

Reviewers' comments:

Reviewer's Responses to Questions

**Comments to the Author**

1. If the authors have adequately addressed your comments raised in a previous round of review and you feel that this manuscript is now acceptable for publication, you may indicate that here to bypass the “Comments to the Author” section, enter your conflict of interest statement in the “Confidential to Editor” section, and submit your "Accept" recommendation.

Reviewer #3: (No Response)

2. Is the manuscript technically sound, and do the data support the conclusions?

Reviewer #3: Yes

3. Has the statistical analysis been performed appropriately and rigorously? 

Reviewer #3: Yes

4. Have the authors made all data underlying the findings in their manuscript fully available?

Reviewer #3: Yes

5. Is the manuscript presented in an intelligible fashion and written in standard English?

Reviewer #3: Yes

6. Review Comments to the Author

Reviewer #3: This manuscript deals with a question that, in my opinion, looks highly relevant for many applications: do negative numbers induce more uncertainty than positive ones? It does so by resorting to different strategies: hardware that allows graded responses in the form of pressure, cognitive modelling with the diffusion model… In general, I like the idea behind this research. However, I had problems to fully understand what the manuscript is describing, in part because the text organization is not clear.

I will describe next a few comments that may help to improve the manuscript:

1. Text organization

I found the Introduction well written and easy to follow. Not being an expert myself, I think that the Introduction contains the basic information needed to understand the purpose and justification of the studies.

The rest of the manuscript is organized in a way that, in my opinion, does not help to understand what is being described:

-The three experiments are described at the same time, which makes it difficult to understand the differences between them and why one needs to conduct three experiments at all. A more traditional organization with each experiment being described (together with its justification) alone would be better, in my opinion. If the experiments have very similar procedure, you could just refer to that explanation in previous sections. Now the motivation to run Experiment 3, for instance, seems completely overlooked (it is mentioned somewhere at the beginning and it seems to never matter again).

-The same happens with the Data Analysis, Drift-Diffusion model, and Results sections. It is hard to keep in mind all the information when one goes through these sections. The Results section, for instance, is just a collection of tables, full of numbers, with little guidance in the text. I would prefer a more traditional organization: experiment-wise, with a description of the data analyses that will be later reported (and explained!) in the Results section. Maybe, too, a final section for the modelling that could include the three experiments, if you want to. Or perhaps the diffusion model can be described in the first Data Analysis section, and then in subsequent subsections of the Results sections for each experiment. Now it is all a bit too mixed up.

2. Theoretical implications.

Admittedly, I am not an expert on numerical cognition. From my reading of the Introduction, I got that there are two competing theories that make different predictions concerning negative numbers. But it is not clear to me whether and how the current studies help in addressing these questions. Would it be possible to connect the results to these theories, perhaps favouring one over the other? (It seems that the diffusion model results indicate that differences between numbers are not an encoding effect)

My impression is that, despite the highly interesting theoretical debate, the results of these experiments are just descriptive: they suggest that there is an uncertainty burden in negative numbers, at least in this task. Thus, we cannot advance too much in theory without further research.

I was curious about a potential extension of these results to different tasks. Would negative numbers produce more uncertain answers in any type of task? For instance: Here, the task is very simple and implies recognizing negative numbers. I don’t know if the result is generalizable to production tasks. What would happen if, facing any numerical task (solving mathematical problems, or just emitting judgments), those responses that are negative produce more uncertainty as well?

3. Minor comments:

-Power analyses. The authors conduct post hoc power analyses. This is not recommended (Althouse, 2021; Hoenig & Heisy, 2001). Post hoc power estimations are *determined* by the p-value. So, once you get a significant result, what is the point in computing power?

What can we do, then? Ideally, power calculations should be conducted *before* data collection (“a priori power analyses”), so that you determine you sample size given an estimated effect size and a desired power level. However, this is not easy to do as the effect size must still be estimated (which is hard to do, as the literature is often biased).

Thus, I assume that you did not conduct power analyses a priori. Then, what you can do is conducting sensitivity analyses. Sensitivity analyses can be conducted a priori or a posteriori, and they reach a compromise between power goals and practical issues. You can use G*Power or any other software to: (1) decide which sample size you are willing to collect (or you have already collected) for practical/ethical/economic reasons, (2) fix a power level (e.g., 80%), and (3) solve for the minimal effect size that you can detect.

If the sensitivity analyses reveal that you can detect reliably (with good power) even small effects, then your study is well powered for those effects. If you can only detect large effects, that means that you have low power for small effects.

Another thing you can do is reporting effect sizes and confidence intervals. Even if the results are significant, a large interval suggests that the study is not informative.

-Lots of t-tests were conducted (e.g. with diffusion model parameters, Table 5) without (as far as I can see) multiple contrasts protection.

- P. 18: “However, the term…” (do you mean “interaction term”?)

References:

Althouse A. D. (2021). Post Hoc Power: Not Empowering, Just Misleading. The Journal of surgical research, 259, A3–A6. https://doi.org/10.1016/j.jss.2019.10.049

Hoenig JM, Heisey DM. The abuse of power: the pervasive fallacy of power calculations for data analysis. Am Stat. 2001; 55:19-24.

7. PLOS authors have the option to publish the peer review history of their article (what does this mean?). If published, this will include your full peer review and any attached files.

Reviewer #3: No

---

## [Author Response · Author response to Decision Letter 3]

11 Jul 2022

#REVIEWER 3

Reviewer #3: This manuscript deals with a question that, in my opinion, looks highly relevant for many applications: do negative numbers induce more uncertainty than positive ones? It does so by resorting to different strategies: hardware that allows graded responses in the form of pressure, cognitive modelling with the diffusion model… In general, I like the idea behind this research. However, I had problems to fully understand what the manuscript is describing, in part because the text organization is not clear.

Response: Thanks for your positive words, we appreciate them. We hope that we address your concerns in this revised manuscript. 

I will describe next a few comments that may help to improve the manuscript:

1. Text organization

I found the Introduction well written and easy to follow. Not being an expert myself, I think that the Introduction contains the basic information needed to understand the purpose and justification of the studies.

The rest of the manuscript is organized in a way that, in my opinion, does not help to understand what is being described:

-The three experiments are described at the same time, which makes it difficult to understand the differences between them and why one needs to conduct three experiments at all. A more traditional organization with each experiment being described (together with its justification) alone would be better, in my opinion. If the experiments have very similar procedure, you could just refer to that explanation in previous sections. Now the motivation to run Experiment 3, for instance, seems completely overlooked (it is mentioned somewhere at the beginning and it seems to never matter again).

Response: In the new manuscript we separated each experiment into different sections. Now the motivation for each one is explained clearly at the beginning of each one. Thus, Exp. 1 is traditional keyboard experiment that allow us to show some confidence predictions from the drift-diffusion model (DDM). Exp. 2 and 3 confirm one of those predictions and replicates the DDM results. Exp. 3 further controls for important methodological aspects.

-The same happens with the Data Analysis, Drift-Diffusion model, and Results sections. It is hard to keep in mind all the information when one goes through these sections. The Results section, for instance, is just a collection of tables, full of numbers, with little guidance in the text. I would prefer a more traditional organization: experiment-wise, with a description of the data analyses that will be later reported (and explained!) in the Results section. 

Response: Now each experiment has its own data analysis and results sections. The drift diffusion is only explained in Exp. 1. By separating each experiment we think that now the explanations are clearer. 

Maybe, too, a final section for the modelling that could include the three experiments, if you want to. Or perhaps the diffusion model can be described in the first Data Analysis section, and then in subsequent subsections of the Results sections for each experiment. Now it is all a bit too mixed up.

Response: We now describe the drift diffusion model in the first Data Analysis (in Exp. 1) and then presented the results for each experiment on their own subsection. 

2. Theoretical implications.

Admittedly, I am not an expert on numerical cognition. From my reading of the Introduction, I got that there are two competing theories that make different predictions concerning negative numbers. But it is not clear to me whether and how the current studies help in addressing these questions. Would it be possible to connect the results to these theories, perhaps favouring one over the other? (It seems that the diffusion model results indicate that differences between numbers are not an encoding effect)

Response: The results indeed point to holistic representations of numbers, not merely encoding. The diffusion model, and the response times patterns in Exp. 1 and 2, surely suggests that. In the discussion section we are now more explicit on saying this (Page 32.) 

My impression is that, despite the highly interesting theoretical debate, the results of these experiments are just descriptive: they suggest that there is an uncertainty burden in negative numbers, at least in this task. Thus, we cannot advance too much in theory without further research.

Response: Indeed, button pressure alone cannot disentangle which theory is correct (at least with our design). We mention this in Page 32. Still response times and the drift diffusion model favor holistic representations and we mention this also in Page 32. However, given the assumptions behind the drift-diffusion model we are still conservative and mention that further research is required.

I was curious about a potential extension of these results to different tasks. Would negative numbers produce more uncertain answers in any type of task? For instance: Here, the task is very simple and implies recognizing negative numbers. I don’t know if the result is generalizable to production tasks. What would happen if, facing any numerical task (solving mathematical problems, or just emitting judgments), those responses that are negative produce more uncertainty as well?

Response: in this new revision the last phrase is suggesting such critical future questions.

3. Minor comments:

-Power analyses. The authors conduct post hoc power analyses. This is not recommended (Althouse, 2021; Hoenig & Heisy, 2001). Post hoc power estimations are *determined* by the p-value. So, once you get a significant result, what is the point in computing power?

What can we do, then? Ideally, power calculations should be conducted *before* data collection (“a priori power analyses”), so that you determine you sample size given an estimated effect size and a desired power level. However, this is not easy to do as the effect size must still be estimated (which is hard to do, as the literature is often biased).

Thus, I assume that you did not conduct power analyses a priori. Then, what you can do is conducting sensitivity analyses. Sensitivity analyses can be conducted a priori or a posteriori, and they reach a compromise between power goals and practical issues. You can use G*Power or any other software to: (1) decide which sample size you are willing to collect (or you have already collected) for practical/ethical/economic reasons, (2) fix a power level (e.g., 80%), and (3) solve for the minimal effect size that you can detect.

If the sensitivity analyses reveal that you can detect reliably (with good power) even small effects, then your study is well powered for those effects. If you can only detect large effects, that means that you have low power for small effects.

Another thing you can do is reporting effect sizes and confidence intervals. Even if the results are significant, a large interval suggests that the study is not informative.

Response: We fully agree with you. In the original manuscript we did not do post-hoc justifications. As the review process advanced, we were trying to understand the requirements of previous reviewers. We agree with you that the sensitivity analysis is a good middle ground. We now provide a sensitivity analysis for Exp. 1. For Exp. 2 we mention that we preregistered a max. sample size and that Exp. 3 also follow that cap.

-Lots of t-tests were conducted (e.g. with diffusion model parameters, Table 5) without (as far as I can see) multiple contrasts protection.

Response: We corrected for multiple comparisons with Holm-Sidak (page 12)

- P. 18: “However, the term…” (do you mean “interaction term”?)

Response: Yes, you are right, we corrected that.

References:

Althouse A. D. (2021). Post Hoc Power: Not Empowering, Just Misleading. The Journal of surgical research, 259, A3–A6. https://doi.org/10.1016/j.jss.2019.10.049

Hoenig JM, Heisey DM. The abuse of power: the pervasive fallacy of power calculations for data analysis. Am Stat. 2001; 55:19-24.

7. PLOS authors have the option to publish the peer review history of their article (what does this mean?). If published, this will include your full peer review and any attached files.

Do you want your identity to be public for this peer review? For information about this choice, including consent withdrawal, please see our Privacy Policy.

Reviewer #3: No

Reviewer #1: 

NOTE: THIS IS FROM THE PREVIOUS ROUND (WITH MINOR EDITS). WE INCLUDE THE BY THE EDITOR’S REQUEST.

1. The authors made several adjustments and additions that strengthen the ms, including explanations of their idiosyncratic terminology (“dummy trials” for filler trials) and references to relevant papers that were previously omitted. However, repeatedly directing readers to Ganor-Stern & Tzelgov (2008) for the component model is still suboptimal because the paper by Huber (their Ref 5) constitutes its more recent development. 

We now extend the component model explanation with the Huber et al neural network (Intro.: Pages 3-4). 

Similarly, the introduction of “button pressure” as the measure of interest (on page 4) is immediately followed by references to “motor metrics”, i.e. kinematic studies, even though the “button pressure” is an isometric assessment without any kinematics involved. In my idiosyncratic view these are suboptimal revisions.

We wanted to show to a general audience that “motor metrics” are used in cognition. We understand that those references were about kinematics. Thus, we now drop those kinematics references to avoid any confusion. 

2. More importantly, the methods descriptions are still not detailed enough to permit replication. Although stimulus ranges are now clear and the recording apparatus for “button pressure” is now explained in more detail, the sensitivity of this device is still unclear. If the range is 0.25 to 0.90, does this mean that 65 levels of pressure were discriminable 

The range 0.25-0.90 is continuous. There are more than 65 discrete levels. 

The actual range from the raw data is 0.025 to 1 (Exp 2) and 0.05 to 1 (Exp. 3). The 0.25 – 0.9 range was after taking averages per numerical distances. In the new manuscript (Data analysis) we decided that it was clearer if we present the ranges from the raw data because the right end is 1 and the left range is close to zero.

while the actual force produced (in Pascal or Newton) remains unknown? 

The sensor detects pressure as a change in resistance. The details of the resistors and Arduino wiring are in the manuscript and supplemental information. The Arduino transforms those changes of resistance into a discrete signal between 0 and 1023. All subjects used the same Arduino setup (including voltage and resistors). Thus, we do not report Pascals or Newtons (the transformation from change in resistance to Newtons or Pascals is not trivial). Still, the change in resistance indeed relates to force; resistance-based pressure is used in many current devices including mobile phones and many portable devices.

Another open issue, of fundamental importance for the later discussion about early attentional vs later conceptual effects: What was the sampling frequency (i.e. the temporal resolution of pressure measurements)?

We now report the sampling frequency of the Arduino board in the manuscript (Apparatus, page 7-8) (approx. 26 hz but see manuscript for details). 

It is not fast enough to fully disentangle early attention and later concepts (holistic magnitudes). 

In the discussion we accept that it could be attention, but it would require some ad-hoc accommodations so that all our tables and results are about attention and not confidence. 

we do think that confidence results speak about concepts, namely the existence of negative holistic magnitudes (see discussion section). However, we are also aware that our paper only brings partial clarity to negative magnitude processing. Our paper is just about confidence.

3. Moreover, despite my extensive queries on this point, I still find only a SINGLE sentence that describes the force data analysis, namely “Data analysis. We used panel linear regressions to analyze the accuracy (linear probability model), response times up to two standard deviations from the mean (i.e., 95% of the trials), and button pressure”. 

We apologize for our lack of clarity; it is not our intention. In the first revision we tried to address all your comments, but we clearly fail. We hope that our responses below clarify our data analysis.

This leaves open fundamental questions such as the time during which force was recorded or integrated, 

We analyzed signals as soon as the numerals appeared on screen and until the subject selected an option and was no longer pressing the force sensor (i.e. force resistance < low_pressure_threshold) (this for all trials). (Data Analysis).

the data filtering or trimming for this inherently noisy signal, 

We did not filter Arduino signals. When the sensor is pressed it sends a reading to Matlab based on the participant’s pressure. In the figure below (see the .docx document rather than the Editorial Manager pdf), each trial has a stereotypical look: a peak. This means that participants had a brief contact with the sensor (here we plot the raw Arduino signal, before any standardization) (see all raw data in Supplemental Figures 18 and 19).

We filtered data on the behavior side. We reported in the manuscript two criteria: response time greater than 2 standard deviations from the mean of all the data and accuracy less than 85%. 

We did detect an anomaly thanks to your comment. We plotted all the raw pressure data and we detected one subject who used the keyboard instead of the Arduino sensors. The pressures of subj. 33443 (Exp. 2) were always at baseline levels (Supp. Figure 18). This subj did respond accurately to the task, meaning that they used the keyboard (we explain this on the participants section of the new manuscript). 

The reason this could happen was that we coded the first experiment (Exp. 1) so that subjects had to use the keyboard. In Exp. 2 we used the same base code (plus the Arduino stuff) and even though we verbally explained to all subjects to use the Arduino sensors, we kept, by mistake, the original instructions of Exp. 1 on screen. Subj. 33443 was the only subject (across Exp. 2 and 3) who mistakenly used the keyboard (see Supplemental Figure 18). We excluded this subject from this revision. Importantly, the main pattern of results did not change. 

or the computation of parameters for analysis, such as average or peak force per trial, or many other candidates. None of this is explained.

For this revision we now provide a concrete example for further clarity (Data Analysis section). Specifically, we used for analysis peak force during the trial, normalized to the max peak force during the whole session as a normalizing constant (separately for the left and right sensor). For instance, if subject X max peak force to the left during his whole session was 984, then all left pressures during the session were divided by 984. We used this metric in the tables. That is, the regression parameters were obtained with standardized peak pressure. We now put explicitly in the regressions the name of the variable followed by “normed”.

4. I am unhappy with various minor aspects of this revision: The authors’ claim to defend their sample size of 50 participants “in studies with similar sample sizes (1,2,4,8)” (p. 6) is misleading readers because only Experiment 2 of ref. 4 has 55 participants, while all others have between 16 and 27. 

We dropped the expression “similar sample size” to the actual range of 16 to 55. Thus, our sample size is on the larger end.

My request to determine power or sensitivity retrospectively was not addressed.

We calculated power, using response times from our participants as it is known in the literature that confidence and response times correlate (Participants: Page 7). Overall, the obtained power is large.

Some wording is still poor: On page 7 “Exp. 1 was run in a 13-inch laptop” (should be “on”),

Solved.

 or “Uncertainty is a more general concept as it is not a conditioned in choice.” (p. 5) is ungrammatical. 

We dropped that expression and rather clarify the distinction of uncertainty and confidence in the model because it is more concrete for the reader: DDM model: Page 13: “It is important to highlight that uncertainty UN is not the same as confidence in the model. UN is the base standard deviation of the inference, while confidence is the cumulative probability that μdv is greater than zero after a choice is made (i.e. after dv arrives at one of the thresholds).” 

Several references contain the superfluous word “internet”.

We did not realize that Mendeley was doing that, sorry.

OTHER IMPORTANT CLARIFICATIONS 

• In Exp. 3, by mistake, we standardized to a value that was not the peak pressure in general but the initial peak value in the Arduino signal greater than a low threshold. We corrected this oversight and that is why some regression estimates of Exp. 3 are different in this current revision. Importantly the overall pattern/direction/significance of the effects is identical, just slightly different estimates (i.e. both constants are correlated because each trial is a time series). This oversight did not affect the conclusions.

The oversight happened because in Exp. 3 we decided to change the serial function used by Matlab to communicate with the Arduino (it will be deprecated; the new one is serialport). Exp. 3 was the experiment that reviewer 1 asked i.e. it was a new experiment and when we changed the code to include the new conditions and Matlab function, the code we wrote marked events slightly differently. During data analyses we missed changing some of the marks and we ended up using, mistakenly, the peak pressure across trials of the first pressure signal sent by the Arduino greater than a low-pressure threshold on each trial. As we mentioned, we corrected this in the current revision.

• In the Table 1 of the previous revision we didn’t include the variable controlling for error feedback. In this revision we fixed that.

• The only tables that include interaction effects are the ones theoretically relevant. In Page 11: “We included interactions in the regressions when it was theoretically relevant. Namely when the dependent variable was response times (distinct slopes by numeral type suggest different mental magnitudes) and pressure (Figure 1, central panel shows distinct slopes for correct and incorrect trials).”

• The manuscript is on psyarxiv with figures on the manuscript (https://psyarxiv.com/sdfb9/)

---

## [Decision Letter · Decision Letter 4]

27 Jul 2022

Reduced choice-confidence in negative numerals

PONE-D-21-25521R4

Dear Dr. Alonso-Diaz,

We’re pleased to inform you that your manuscript has been judged scientifically suitable for publication and will be formally accepted for publication once it meets all outstanding technical requirements.

Kind regards,

Federico Giove, PhD

Academic Editor

PLOS ONE

Additional Editor Comments (optional):

Reviewers' comments:

Reviewer's Responses to Questions

**Comments to the Author**

1. If the authors have adequately addressed your comments raised in a previous round of review and you feel that this manuscript is now acceptable for publication, you may indicate that here to bypass the “Comments to the Author” section, enter your conflict of interest statement in the “Confidential to Editor” section, and submit your "Accept" recommendation.

Reviewer #3: (No Response)

2. Is the manuscript technically sound, and do the data support the conclusions?

Reviewer #3: Yes

3. Has the statistical analysis been performed appropriately and rigorously? 

Reviewer #3: Yes

4. Have the authors made all data underlying the findings in their manuscript fully available?

Reviewer #3: Yes

5. Is the manuscript presented in an intelligible fashion and written in standard English?

Reviewer #3: Yes

6. Review Comments to the Author

Reviewer #3: I think the revised manuscript addresses my previous comments. Specifically:

-The text organization is much better, which makes it easier to read.

-The purpose of the experiments is clearer.

-The connection to previous literature/hypotheses is also better explained.

Still, I am not sure that I fully understand the statistical analyses and some other technical details about the pressure measure (that other reviewers seem to find important), so although I think that they are sound, please take my comments with a grain of salt.

Minor comments:

-When reporting sensitivity analyses, you fix alpha (the error rate) to 0.05, rather than “p” as you describe: “we did a sensitivity analysis to check for the required effect size given an 80% power, p<0.05, …”. The p-value is a statistic you obtain from the data and it is not fixed beforehand.

7. PLOS authors have the option to publish the peer review history of their article (what does this mean?). If published, this will include your full peer review and any attached files.

Reviewer #3: No

---

## [Editor Report · Acceptance letter]

1 Aug 2022

PONE-D-21-25521R4 

Reduced choice-confidence in negative numerals 

Dear Dr. Alonso-Diaz:

I'm pleased to inform you that your manuscript has been deemed suitable for publication in PLOS ONE. Congratulations! Your manuscript is now with our production department. 

Kind regards, 

on behalf of

Dr. Federico Giove 

Academic Editor

PLOS ONE